# Hebbian Learning based Orthogonal Projection for Continual Learning of Spiking Neural Networks

**Mingqing Xiao[1], Qingyan Meng[2,3], Zongpeng Zhang[4], Di He[1,5], Zhouchen Lin[1,5,6*]**
[1]National Key Lab of General AI, School of Intelligence Science and Technology, Peking University
[2]The Chinese University of Hong Kong, Shenzhen
[3]Shenzhen Research Institute of Big Data
[4]Department of Biostatistics, School of Public Health, Peking University
[5]Institute for Artificial Intelligence, Peking University
[6]Peng Cheng Laboratory

## Abstract

Neuromorphic computing with spiking neural networks is promising for energy-efficient artificial intelligence (AI) applications. However, different from humans who continually learn different tasks in a lifetime, neural network models suffer from catastrophic forgetting. How could neuronal operations solve this problem is an important question for AI and neuroscience. Many previous studies draw inspiration from observed neuroscience phenomena and propose episodic replay or synaptic metaplasticity, but they are not guaranteed to explicitly preserve knowledge for neuron populations. Other works focus on machine learning methods with more mathematical grounding, e.g., orthogonal projection on high dimensional spaces, but there is no neural correspondence for neuromorphic computing. In this work, we develop a new method with neuronal operations based on lateral connections and Hebbian learning, which can protect knowledge by projecting activity traces of neurons into an orthogonal subspace so that synaptic weight update will not interfere with old tasks. We show that Hebbian and anti-Hebbian learning on recurrent lateral connections can effectively extract the principal subspace of neural activities and enable orthogonal projection. This provides new insights into how neural circuits and Hebbian learning can help continual learning, and also how the concept of orthogonal projection can be realized in neuronal systems. Our method is also flexible to utilize arbitrary training methods based on presynaptic activities/traces. Experiments show that our method consistently solves forgetting for spiking neural networks with nearly zero forgetting under various supervised training methods with different error propagation approaches, and outperforms previous approaches under various settings. Our method can pave a solid path for building continual neuromorphic computing systems. The code is available at `https://github.com/pkuxmq/HLOP-SNN`.

## 1 Introduction

Brain-inspired neuromorphic computing, e.g., biologically plausible spiking neural networks (SNNs), has attracted tremendous attention (Roy et al., 2019) and achieved promising results with deep architectures and gradient-based methods recently (Shrestha & Orchard, 2018; Zheng et al., 2021; Fang et al., 2021; Yin et al., 2021; Meng et al., 2022). The neuromorphic computing systems imitate biological neurons with parallel in-memory and event-driven neuronal computation and can be efficiently deployed on neuromorphic hardware for energy-efficient applications (Akopyan et al., 2015; Davies et al., 2018; Pei et al., 2019; Woźniak et al., 2020; Rao et al., 2022). SNNs are also believed as one of important approaches toward artificial general intelligence (Pei et al., 2019).

However, neural network models typically suffer from catastrophic forgetting of old tasks when learning new ones (French, 1999; Goodfellow et al., 2013; Kirkpatrick et al., 2017), which is sig-

---

*corresponding author

nificantly different from the biological systems of human brains that can build up knowledge in the whole lifetime. This is because the weights in the network are updated for new tasks and are usually not guaranteed to preserve the knowledge of old tasks during learning. To establish human-like life-long learning in dynamically changing environments, developing neuromorphic computing systems of SNNs with continual learning (CL) ability is important for artificial intelligence.

To deal with this problem, biological underpinnings are investigated (Kudithipudi et al., 2022), and various machine learning methods are proposed. Most works can be categorized into two classes.

The first category draws inspiration from observed neuroscience phenomena and proposes methods such as replay (Rebuffi et al., 2017; Shin et al., 2017; Rolnick et al., 2019; van de Ven et al., 2020) or regularization (Kirkpatrick et al., 2017; Zenke et al., 2017; Aljundi et al., 2018; Laborieux et al., 2021). For example, replay methods are inspired by the episodic replay phenomenon during sleep or rest, and regularization is based on the metaplasticity of synapses, i.e., the ease with which a synapse can be modified depends on the history of synaptic modifications and neural activities (Kudithipudi et al., 2022). However, replay methods only implicitly protect knowledge by retraining and require storing a lot of old input samples for frequent replay, and metaplasticity methods only focus on regularization for each individual synapse, which is not guaranteed to preserve knowledge considering the high dimensional transformation of interactive neuron populations. It is also unclear how other biological rules such as Hebbian learning can systematically support continual learning.

The second category focuses on developing pure machine learning methods with complex computation, e.g., pursuing better knowledge preservation considering projection on high dimensional spaces (He & Jaeger, 2018; Zeng et al., 2019; Saha et al., 2021), but they require operations such as calculating the inverse of matrices or singular value decomposition (SVD), which may not be implemented by neuronal operations. Considering neuromorphic computing systems of SNNs, it is important to investigate how neural computation can deal with the problem.

In this work, we develop a new method, Hebbian learning based orthogonal projection (HLOP), for task- and domain-incremental continual learning of neuromorphic computing systems, which is based on lateral connections as well as Hebbian and anti-Hebbian learning. The core idea is to use lateral neural computation with Hebbian learning to extract principle subspaces of neuronal activities from streaming data and project activity traces for synaptic weight update into an orthogonal subspace so that the learned abilities are not influenced much when learning new knowledge. Such a way of learning is flexible to utilize arbitrary training methods based on presynaptic activity traces.

Our method can explicitly preserve old knowledge by neural computation without frequent replay, and impose a more systematic constraint for weight update considering neuron populations instead of individual synapses, compared with replay and metaplasticity methods. Meanwhile, our method focuses on knowledge protection during the instantaneous learning of new tasks and is compatible with post-processing methods such as episodic replay. Compared with previous projection methods, our method enables a better, unbiased construction of the projection space based on online learning from streaming large data instead of a small batch of data (Saha et al., 2021), which leads to better performance, and our work is also the first to show how the concept of projection with more mathematical guarantees can be realized with pure neuronal operations.

Experiments on continual learning of spiking neural networks under various settings and training methods with different error propagation approaches consistently show the superior performance of HLOP with nearly zero forgetting, outperforming previous methods. Our results indicate that lateral circuits can play a substantial role in continual learning, providing new insights into how neural circuits and Hebbian learning can support the advanced abilities of neural systems. With purely neuronal operations, HLOP can also pave paths for continual neuromorphic computing systems.

## 2 RELATED WORK

Many previous works explore continual learning of neural networks. Most methods focus on artificial neural networks (ANNs) and only few replay or metaplasticity/local learning works (Tadros et al., 2022; Soures et al., 2021; Panda et al., 2017; Wu et al., 2022) consider SNNs. Methods for fixed-capacity networks can be mainly categorized as replay, regularization-based, and parameter isolation (De Lange et al., 2021). Replay methods will replay stored samples (Rebuffi et al., 2017; Rolnick et al., 2019) or generated pseudo-samples (Shin et al., 2017; van de Ven et al., 2020) during or after learning a new task to serve as rehearsals or restrictions (Lopez-Paz & Ranzato,

2017; Chaudhry et al., 2019), which are inspired by the episodic replay during sleep or rest in the brain (Kudithipudi et al., 2022). Regularization-based methods are mostly inspired by the meta-plasticity of synapses in neuroscience, and regularize the update of each synaptic weight based on the importance estimated by various methods (Kirkpatrick et al., 2017; Zenke et al., 2017; Aljundi et al., 2018; Laborieux et al., 2021; Soures et al., 2021) or knowledge distillation (Li & Hoiem, 2017). Parameter isolation methods separate parameters for different tasks, e.g., with pruning and mask (Mallya & Lazebnik, 2018) or hard attention (Serra et al., 2018). There are also expansion methods (Rusu et al., 2016; Yoon et al., 2018) to enlarge network capacity for every new task.

Orthogonal projection methods take a further step of regularization to consider the projection of gradient so that the update of weight is constrained to an orthogonal direction considering the high-dimensional linear transformation. Farajtabar et al. (Farajtabar et al., 2020) store previous gradients and project gradients into the direction orthogonal to old ones. He & Jaeger (2018), Zeng et al. (2019) and Saha et al. (2021) project gradients into the orthogonal subspace of the input space for more guarantee and better performance. Lin et al. (2022) improve Saha et al. (2021) with trust region projection considering similarity between tasks. However, these methods only estimate the subspace with a small batch of data due to the large costs of SVD, and there is no neural correspondence. We generalize the thought of projection to neuronal operations under streaming data.

Meanwhile, continual learning can be classified into three fundamental types: task-incremental, domain-incremental, and class-incremental (van de Ven et al., 2022). Task-CL requires task ID at training and testing, domain-CL does not need task ID, while class-CL requires comparing new tasks with old tasks and inferring context without task ID. Following previous gradient projection methods, we mainly consider task- and domain-incremental settings. As for class-CL, it inevitably requires some kind of replay of old experiences (via samples or other techniques) for good performance since it expects explicit comparison between new classes and old ones (van de Ven et al., 2020). Our method may be combined with replay methods to first update with projection and then learn with replay, or with context-dependent processing modules similar to biological systems (Zeng et al., 2019; Kudithipudi et al., 2022), e.g., task classifiers. This is not the focus of this paper.

## 3 PRELIMINARIES

### 3.1 SPIKING NEURAL NETWORK

We consider SNNs with the commonly used leaky integrate and fire (LIF) neuron model as in previous works (Wu et al., 2018; Xiao et al., 2022). Each spiking neuron maintains a membrane potential $u$ to integrate input spike trains and generates a spike once $u$ reaches a threshold. The dynamics of the membrane potential are $\tau_m \frac{du}{dt} = -(u - u_{rest}) + R \cdot I(t)$ for $u < V_{th}$, where $I$ is the input current, $V_{th}$ is the threshold, and $R$ and $\tau_m$ are resistance and time constant, respectively. When $u$ reaches $V_{th}$ at time $t^f$, a spike is generated and $u$ is reset to the resting potential $u = u_{rest}$. The output spike train is defined using the Dirac delta function: $s(t) = \sum_{t^f} \delta(t - t^f)$. Consider the simple current model $I_i(t) = \sum_j w_{ij} s_j(t) + b_i$, where $i$ represents the $i$-th neuron, $w_{ij}$ is the weight from neuron $j$ to neuron $i$ and $b_i$ is a bias, then the discrete computational form is described as:

$$\begin{cases} u_i[t+1] = \lambda(u_i[t] - V_{th} s_i[t]) + \sum_j w_{ij} s_j[t] + b_i, \\ s_i[t+1] = H(u_i[t+1] - V_{th}), \end{cases} \tag{1}$$

where $H(x)$ is the Heaviside step function, $s_i[t]$ is the spike train of neuron $i$ at discrete time step $t$, and $\lambda < 1$ is a leaky term based on $\tau_m$ and the discrete step size. The constant $R$, $\tau_m$, and time step size are absorbed into the weights and bias. The reset operation is implemented by subtraction.

Different from ANNs, the supervised learning of SNNs is hard due to the non-differentiability of spiking neurons, and different methods are proposed to tackle the problem. One kind of training approach is based on the explicit encoding of spike trains, such as (weighted) firing rate (Wu et al., 2021; Xiao et al., 2021; Meng et al., 2022) or spiking time (Mostafa, 2017), and calculates gradients through analytical transformations between the spike representation. Another kind is spike-based approaches by backpropagation through time (BPTT) and surrogate gradients (SG) without the explicit coding scheme (Shrestha & Orchard, 2018; Bellec et al., 2018; Wu et al., 2018; Zheng et al., 2021; Li et al., 2021; Fang et al., 2021; Yin et al., 2021). Some works further improve BPTT for temporally online learning that is more consistent with biological learning and friendly for on-chip training (Bellec et al., 2020; Xiao et al., 2022; Meng et al., 2023). We will consider these three kinds of training methods and more details can be found in Appendix A.

## 3.2 ORTHOGONAL GRADIENT PROJECTION

Orthogonal gradient projection methods protect knowledge for each neural network layer. For feed-forward synaptic weights $\mathbf{W} \in \mathbb{R}^{m \times n}$ between two layers, suppose that the presynaptic input vectors $\mathbf{x}^{old} \in \mathbb{R}^n$ in previous tasks span a subspace $\mathbf{X}$, the goal is to make the updates $\Delta \mathbf{W}^P$ orthogonal to the subspace $\mathbf{X}$ so that $\forall \mathbf{x}^{old} \in \mathbf{X}, \Delta \mathbf{W}^P \mathbf{x}^{old} = \mathbf{0}$. This ensures that the weight update does not interfere with old tasks, since $(\mathbf{W} + \Delta \mathbf{W}^P)\mathbf{x}^{old} = \mathbf{W} \mathbf{x}^{old}$. If we calculate a projection matrix $\mathbf{P}$ to the subspace orthogonal to the principal subspace of $\mathbf{X}$, then gradients can be projected as $\Delta \mathbf{W}^P = \Delta \mathbf{W} \mathbf{P}^\top$. Previous works leverage different methods to calculate $\mathbf{P}$. For example, Zeng et al. (2019) estimate it as $\mathbf{P} = \mathbf{I} - \mathbf{A}(\mathbf{A}^\top \mathbf{A} + \alpha \mathbf{I})^{-1} \mathbf{A}$ (where $\mathbf{A}$ represents all previous inputs) with the recursive least square algorithm. Saha et al. (2021) instead calculate $\mathbf{P} = \mathbf{I} - \mathbf{M}^\top \mathbf{M}$, where $\mathbf{M} \in \mathbb{R}^{k \times n}$ denotes the matrix of top $k$ principal components of $\mathbf{X}$ calculated by SVD with a small batch of data and $\mathbf{M}^\top \mathbf{M}$ is the projection matrix to the principal subspace of $\mathbf{X}$. However, these methods cannot be implemented by neuronal operations for neuromorphic computing, and they only mainly estimate the projection matrix with a small batch of data, which can be biased.

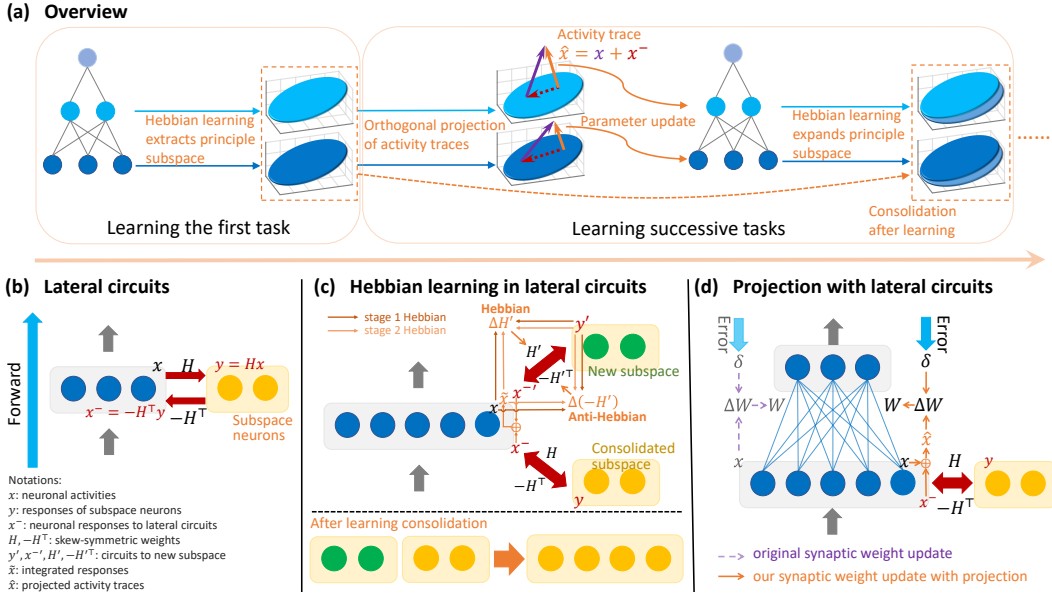

Figure 1: Illustration of the proposed Hebbian learning based orthogonal projection (HLOP). (a) Overview of HLOP for continual learning. Hebbian learning extracts the principal subspace of neuronal activities to support orthogonal projection. For successive tasks, orthogonal projection is based on the consolidated subspace, and a new subspace is constructed by Hebbian learning, which is merged for consolidation after learning. (b) Illustration of lateral circuits with skew-symmetric connection weights. (c) Hebbian learning in lateral circuits for construction of new subspaces for new tasks. (d) Orthogonal projection based on recurrent lateral connections. The presynaptic activity traces $\mathbf{x}$, whose definition depends on training algorithms, are modified by signals from lateral connections. Synaptic weights $\mathbf{W}$ are updated based on the projected traces.

## 4 METHODS

### 4.1 HEBBIAN LEARNING BASED ORTHOGONAL PROJECTION

We propose Hebbian learning based orthogonal projection (HLOP) to achieve orthogonal projection with lateral neural connections and Hebbian learning, and the sketch is presented in Fig. 1. The major conception is to leverage Hebbian learning in a lateral circuit to extract the principal subspace of neural activities in current tasks, which enables the lateral circuit to project activity traces (traces recording activation information used for synaptic update) into the orthogonal subspace, as illustrated in Fig. 1(a). To this end, different from the common feedforward networks, our models consider recurrent lateral connections to a set of "subspace neurons" for each layer (Fig. 1(b)). Lateral connections will not influence feedforward propagation and are only for weight updates.

Recall that the thought of orthogonal projection is to project gradients with $\mathbf{P} = \mathbf{I} - \mathbf{M}^\top \mathbf{M}$ (Section 3.2). Since existing supervised learning methods calculate the weight update as $\Delta \mathbf{W} = \boldsymbol{\delta} \mathbf{x}^\top$, where $\boldsymbol{\delta}$ is the error signal and $\mathbf{x}$ is the activity trace of neurons whose definition depends on training algorithms, we can correspondingly obtain the projected update by $\Delta \mathbf{W}^P = \boldsymbol{\delta} \left( \mathbf{x} - \mathbf{M}^\top \mathbf{M} \mathbf{x} \right)^\top$. The above calculation only requires linear modification on presynaptic activity traces $\mathbf{x}$ which can be implemented by recurrent lateral connections with skew-symmetric synaptic weights, as shown in Fig. 1(b,d). Specifically, the lateral circuit first propagates $\mathbf{x}$ to "subspace neurons" with connections $\mathbf{H}$ as $\mathbf{y} = \mathbf{H} \mathbf{x}$, and then recurrently propagates $\mathbf{y}$ with connections $-\mathbf{H}$ for the response $\mathbf{x}^- = -\mathbf{H}^\top \mathbf{y}$. The activity trace will be updated as $\hat{\mathbf{x}} = \mathbf{x} + \mathbf{x}^- = \mathbf{x} - \mathbf{H}^\top \mathbf{H} \mathbf{x}$. So as long as the lateral connections $\mathbf{H}$ equal or behave similarly to the principal component matrix $\mathbf{M}$, the neural circuits can implement orthogonal projection.

Actually, $\mathbf{H}$ can be updated by Hebbian learning to fulfill the requirements, as shown in Fig. 1(c). Hebbian-type learning has long been regarded as the basic learning method in neural systems (Hebb, 2005) and has shown the ability to extract principal components of streaming inputs (Oja, 1982; 1989). Particularly, Oja's rule (Oja, 1982) was first proposed to update weights $\mathbf{h}$ of input neurons $\mathbf{x}$ to one-unit output neuron $y$ as $\mathbf{h}_t = \mathbf{h}_{t-1} + \eta_t y_t (\mathbf{x}_t - y_t \mathbf{h}_{t-1})$ so that $\mathbf{h}_t$ will converge to the principal component of streaming inputs, which can achieve the state-of-the-art convergence rate in the streaming principal component analysis problem (Chou & Wang, 2020). Several methods extend original Oja's rule to top $k$ principal components (Oja, 1989; Diamantaras & Kung, 1996; Pehlevan et al., 2015), and we focus on the subspace algorithm that updates weights as $\Delta \mathbf{H} = \eta \left( \mathbf{y} \mathbf{x}^\top - \mathbf{y} \mathbf{y}^\top \mathbf{H} \right)$, which enables weights to converge to a dominant principal subspace (Yan et al., 1994). We propose to again leverage the recurrent lateral connections for Hebbian learning. The neurons first propagate signals to "subspace neurons" as $\mathbf{y} = \mathbf{H} \mathbf{x}$, and "subspace neurons" transmit information through the recurrent connections as $\mathbf{x}^- = -\mathbf{H}^\top \mathbf{y}$. The connection weights $\mathbf{H}$ are updated by $\Delta \mathbf{H} = \mathbf{y} \mathbf{x}^\top + \mathbf{y} \mathbf{x}^{-\top}$ with two-stage Hebbian rules considering presynaptic activities and postsynaptic responses. The skew-symmetric weights share symmetric but opposite update directions, which corresponds to the Hebbian and anti-Hebbian learning, respectively. The above condition is the construction of the first subspace neurons. If we further consider learning new subspace neurons with connections $\mathbf{H}'$ for new tasks with existing consolidated subspace neurons, we can only update the connections to new neurons by the similar Hebbian-type learning with presynaptic activities and integrated postsynaptic responses $\tilde{\mathbf{x}} = \mathbf{x}^- + \mathbf{x}^{-\prime}$ from both new and consolidated subspace neurons, i.e., $\Delta \mathbf{H}' = \mathbf{y}' \mathbf{x}^\top + \mathbf{y}' \tilde{\mathbf{x}}^\top$, as shown in Fig. 1(c). This enables the extraction of new principal subspaces not included in the existing subspace neurons unbiasedly from streaming large data. More details are in Appendix C.

HLOP is neuromorphic-friendly as it only requires neuronal operations with neural circuits. At the same time, HLOP may better leverage the asynchronous parallel advantage of neuromorphic computing since the projection and Hebbian learning can be parallel with the forward and error propagation of the feedforward network. The proposed HLOP also has wide adaptability to the combination with training methods based on presynaptic activity traces and other post-processing methods. As HLOP mainly focuses on the constraint of weight update direction during the instantaneous learning of new tasks, it can also be combined with post-processing methods such as episodic replay after the learning to improve knowledge transfer between different tasks.

## 4.2 COMBINATION WITH SNN TRAINING

The basic thought of HLOP is to modify presynaptic traces, and thus HLOP can be flexibly plugged into arbitrary training methods that are based on the traces, such as various SNN training methods.

As introduced in Section 3.1, we will consider different SNN training methods. The definition of activity traces varies for diverse training methods, and HLOP is adapted differently. For methods with explicit coding schemes, activity traces correspond to specified spike representations of spike trains (e.g., weighted firing rates). HLOP acts on the presynaptic spike representation. For BPTT with SG methods, activity traces correspond to spikes at each time step. HLOP can recursively act on presynaptic spike signals at all time steps. For online methods with eligibility traces, activity traces correspond to eligibility traces. HLOP can act on traces at all time steps, which can be implemented by modifying eligibility traces with neuronal responses to lateral circuits.

We illustrate the combination between HLOP and eligibility trace based methods in Fig. 2 (more details about other training methods can be found in Appendix A). Such temporarily online methods

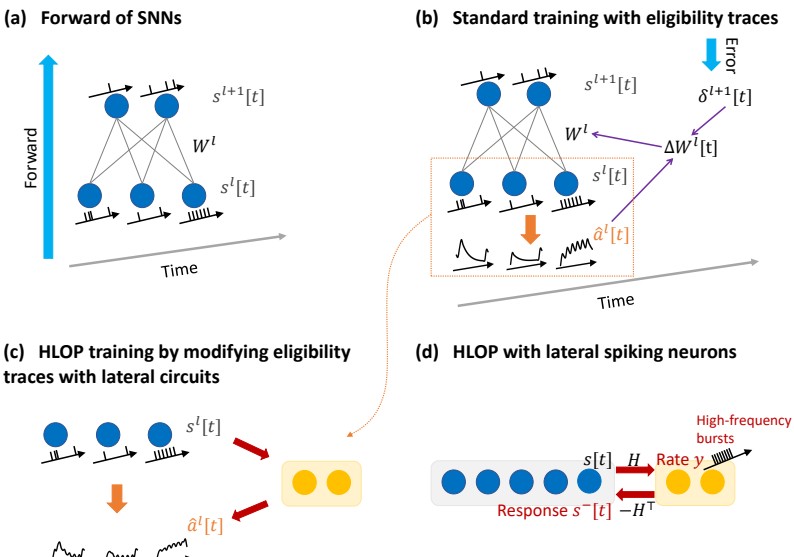

Figure 2: Illustration of combining HLOP with SNN training. (a) Illustration of spiking neural networks with temporal spike trains. (b) SNN training methods based on eligibility traces and online calculation through time (Bellec et al., 2020; Xiao et al., 2022). (c) Applying HLOP by modifying traces with lateral circuits. The eligibility traces correspond to activity traces in the illustration of HLOP and are modified for synaptic weight update. Neuronal traces will consider both spike responses for feedforward inputs and responses for recurrent lateral circuits. (d) Illustration of HLOP with lateral spiking neurons based on the rate coding of high-frequency bursts.

may better suit on-chip training of SNNs, and HLOP can be integrated with online modification of traces by taking additional neuronal response to lateral circuits into account. In experiments, we will consider DSR (Meng et al., 2022), BPTT with SG (Wu et al., 2018; Shrestha & Orchard, 2018), and OTTT (Xiao et al., 2022) as the representative of the three kinds of methods.

The original HLOP leverages linear neurons in lateral circuits. Considering neuromorphic computing systems, this may be supported by some hybrid hardware (Pei et al., 2019). For more general systems, it is important to investigate if HLOP can be implemented by the same spiking neurons as well. To this end, we also propose HLOP with lateral spiking neurons based on the rate coding of high-frequency bursts. As shown in Fig. 2(d), spiking neurons in lateral circuits will generate high-frequency bursts, and the responses are based on spiking rates. We simulate this by quantizing the output of subspace neurons in experiments, i.e., the output $y$ is quantized as $\hat{y} = scale \times \frac{\left[\frac{\text{clamp}(y, -scale, scale)}{scale} \times T_l\right]}{T_l}$, where $scale$ is taken as 20 and $T_l$ is the time steps of lateral spiking neurons as specified in experiments (see Appendix D for more details).

## 5 EXPERIMENTS

We conduct experiments for comprehensive evaluation under different training settings, input domain settings, datasets, and network architectures. We consider two metrics to evaluate continual learning: Accuracy and Backward Transfer (BWT). Accuracy measures the performance on one task, while BWT indicates the influence of learning new tasks on the old ones, which is defined as $\text{BWT}_{k,i} = \text{Accuracy}_{k,i} - \text{Accuracy}_{i,i}$, where $\text{Accuracy}_{k,i}$ is the accuracy on the $i$-th task after learning $k$ tasks ($k \geq i$). The average accuracy after task $k$ is the average of these $k$ tasks, while the average BWT is the average of previous $k - 1$ tasks. Experimental details are in Appendix E.

### 5.1 RESULTS OF DIFFERENT SNN TRAINING METHODS

We first evaluate the effectiveness of HLOP for different SNN training methods on the PMNIST dataset under the online and domain-incremental setting. As shown in Table 1, HLOP significantly improves the results by overcoming catastrophic forgetting under all training methods. The vanilla baseline (i.e., direct training for sequential tasks) suffers from significant forgetting ($> 20\%$) that leads to poor accuracy, while HLOP can achieve almost no forgetting and behave similarly for different training methods. In the following, we leverage DSR as the representative for more settings.

Table 1: Results of average accuracy (ACC) and average backward transfer (BWT) for SNNs on PMNIST under different training methods.

| SNN Training Method | Method | ACC (%) | BWT (%) |
|---|---|---|---|
| DSR | Baseline | 70.61 | $-28.88$ |
| DSR | HLOP | 95.15 | $-1.30$ |
| BPTT with SG | Baseline | 71.17 | $-28.23$ |
| BPTT with SG | HLOP | 95.08 | $-1.54$ |
| OTTT | Baseline | 74.30 | $-24.86$ |
| OTTT | HLOP | 94.76 | $-1.56$ |

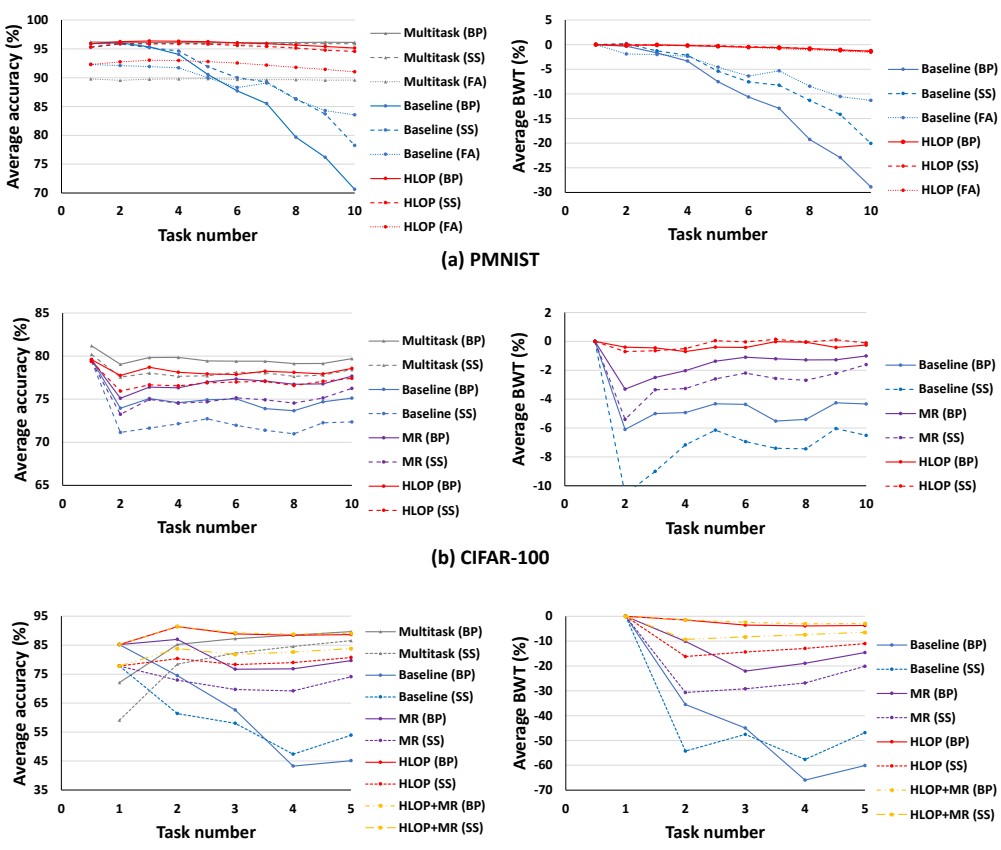

Figure 3: Continual learning results under different settings. "Multitask" does not adhere to continual learning and can be viewed as an upper bound. "BP" (backpropagation), "FA" (feedback alignment), "SS" (sign symmetric) denote different error propagation methods. "MR" denotes memory replay. (a) Average accuracy and backward transfer (BWT) results on PMNIST under the online setting. (b) Average accuracy and BWT results on 10-split CIFAR-100. (c) Average accuracy and BWT results on 5-Datasets (CIFAR-10, MNIST, SVHN, FashionMNIST, and notMNIST).

## 5.2 RESULTS UNDER DIFFERENT DATASETS AND SETTINGS

Then we consider three different settings similar to Saha et al. (2021): (1) continual learning on PMNIST with fully connected networks and single-head classifier (i.e., domain-incremental) under the online setup (i.e., each sample only appears once); (2) continual learning on 10-split CIFAR-100 with convolutional network and multi-head classifiers (i.e., task-incremental); (3) continual learning on 5-Datasets (i.e., the sequential learning of CIFAR-10, MNIST, SVHN, FashionMNIST, and notMNIST) with deeper ResNet-18 networks and multi-head classifiers (i.e., task-incremental).

These settings cover comprehensive evaluation with similar (CIFAR-100) or distinct (PMNIST, 5-Datasets) input domains, task- and domain-incremental settings, online or multi-epochs training,

simple to complex datasets, and shallow fully connected to deeper convolutional architectures. We also consider different error propagation methods including backpropagation (BP), feedback alignment (FA) (Lillicrap et al., 2016), and sign symmetric (SS) (Xiao et al., 2018). Since BP is often regarded as biologically implausible due to the "weight transport" problem (Lillicrap et al., 2016)[1], FA and SS can handle the problem as more biologically plausible alternatives by relaxing feedback weights as random weights or only sharing the sign of forward weights (see Appendix B for more details). They differ from BP in the way to propagate error signals while still calculating gradients based on presynaptic activity traces and errors, so they are naturally compatible with HLOP. Since FA does not work well for large convolutional networks (Xiao et al., 2018), we only consider FA in the PMNIST task with fully connected layers and include SS in all settings.

As shown in Fig. 3, HLOP consistently solves catastrophic forgetting of SNNs under different settings as well as different error propagation methods. While in the baseline setting the forgetting differs for different error propagation methods, it is similar when HLOP is applied, demonstrating the consistent ability of HLOP to overcome forgetting. Compared with memory replay (MR) methods, HLOP achieves higher accuracy and less forgetting, and HLOP can also be combined with memory replay to further improve performance.

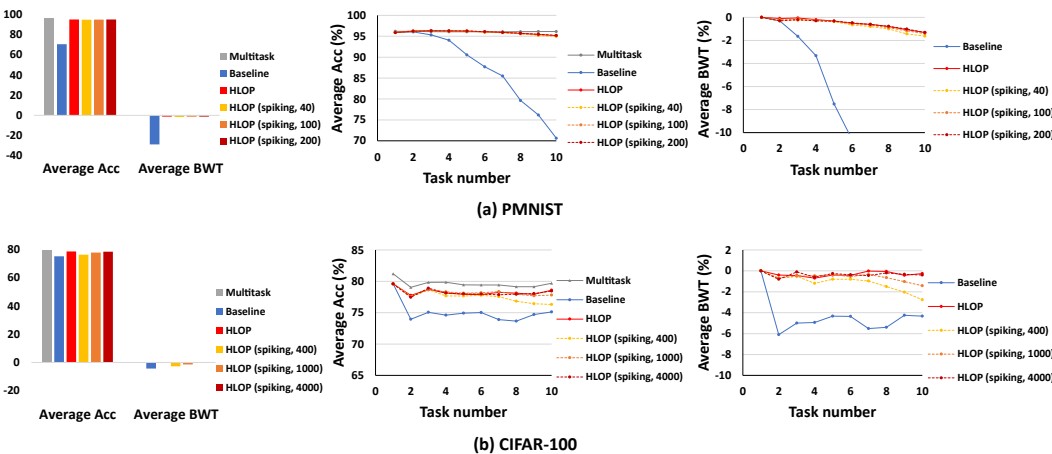

Figure 4: Continual learning results of HLOP with lateral spiking neurons. "Multitask" does not adhere to continual learning and can be viewed as an upper bound. "(spiking, $T_l$)" denotes lateral spiking neurons with $T_l$ time steps for discrete simulation of bursts. (a) Average accuracy and backward transfer (BWT) results on PMNIST under different time steps for HLOP (spiking). (b) Average accuracy and BWT results on 10-split CIFAR-100 under different time steps for HLOP (spiking).

### 5.3 RESULTS OF HLOP WITH LATERAL SPIKING NEURONS

We verify the effectiveness of HLOP with lateral spiking neurons and compare the results with the original linear HLOP in Fig. 4. As shown in the results, HLOP (spiking) also effectively deals with forgetting. And on PMNIST, HLOP (spiking) can achieve similar performance with a small number of time steps (e.g., 40). It shows that we can effectively leverage spiking neurons to implement the whole neural network model and achieve good performance for continual learning.

### 5.4 COMPARISON WITH OTHER CONTINUAL LEARNING METHODS

We compare the performance with other representative continual learning methods. We compare HLOP to the baseline, the small memory replay (MR) method, the regularization-based method EWC (Kirkpatrick et al., 2017), the parameter isolation method HAT (Serra et al., 2018), and the gradient projection-based method GPM (Saha et al., 2021). We consider Multitask as an upper bound, in which all tasks are available and learned together. Experiments on task-incremental 20-split miniImageNet are also compared following Saha et al. (2021). We also provide results for the domain-CL setting on 5-Datasets and comparison results between ANNs and SNNs in Appendix G.

---

[1]That means the inverse connection weights between neurons should be exactly symmetric to the forward ones, which is hard to realize for unidirectional synapses in biological systems and neuromorphic hardware.

Table 2: Comparison results with other continual learning methods under different settings.

| Method | PMNIST[1] | | 10-split CIFAR-100 | | miniImageNet | | 5-Datasets | |
|---|---|---|---|---|---|---|---|---|
| | ACC (%) | BWT (%) | ACC (%) | BWT (%) | ACC (%) | BWT (%) | ACC (%) | BWT (%) |
| *Multitask*[2] | *96.15* | */* | *79.70* | */* | *79.05* | */* | *89.67* | */* |
| Baseline | 70.61 | −28.88 | 75.13 | −4.33 | 40.56 | −32.42 | 45.12 | −60.12 |
| MR | 92.53 | −4.73 | 77.67 | −1.01 | 58.15 | −8.71 | 79.66 | −14.61 |
| EWC | 91.45 | −3.20 | 73.75 | −4.89 | 47.29 | −26.77 | 57.06 | −44.55 |
| HAT | N.A. | N.A. | 73.67 | **−0.13** | 50.11 | −7.63 | 72.72 | −22.90 |
| GPM | 94.80 | −1.62 | 77.48 | −1.37 | 63.07 | −2.57 | 79.70 | −15.52 |
| **HLOP (ours)** | **95.15** | **−1.30** | **78.58** | −0.26 | **63.40** | **−0.48** | **88.65** | **−3.71** |

[1] Experiments are under the online setting, i.e., each sample only appears once (except for the Memory Replay method), and the domain-incremental setting.
[2] Multitask does not adhere to the continual learning setting and can be viewed as an upper bound.

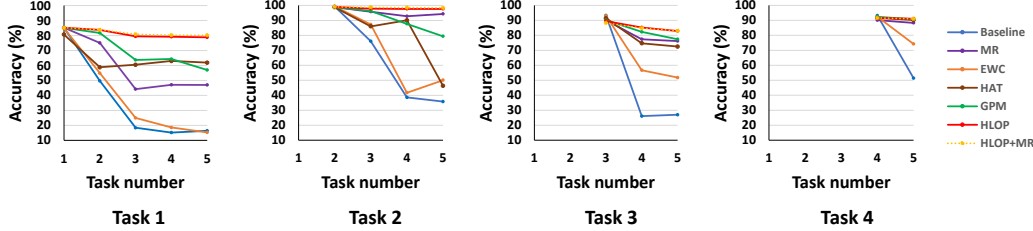

Figure 5: Continual learning results of different methods on 5-Datasets for each task after learning successive ones.

As shown in Table 2, HLOP consistently outperforms previous continual learning approaches under various settings considering both accuracy and backward transfer. The comparison results of accuracy for each task after learning successive ones by different methods on 5-Datasets is illustrated in Fig. 5, showing that HLOP successfully solves forgetting for each task and outperforms previous approaches. Compared with replay and regularization methods, HLOP enables less forgetting and higher accuracy due to the better constraint of weight update direction. Compared with the parameter isolation method, HLOP achieves higher accuracy due to the flexibility of network constraints with high-dimensional subspaces rather than explicit neuronal separation. Compared with the previous gradient projection-based method, HLOP achieves better performance since HLOP has a better construction of projection space from streaming data, especially on the 5-Datasets with distinct input distributions. Moreover, HLOP is based on purely neuronal operations friendly for neuromorphic hardware. These results show the superior performance of HLOP and indicate the potential for building high-performance continual neuromorphic computing systems.

# 6 CONCLUSION

In this work, we propose a new method based on lateral connections and Hebbian learning for continual learning of SNNs. Our study demonstrates that lateral neural circuits and Hebbian learning can systematically provide strong continual learning ability by extracting principle subspaces of neural activities and modifying presynaptic activity traces for projection. This sheds light on how neural circuits and biological learning rules can support the advanced abilities of neuromorphic computing systems, which is rarely considered by popular neural network models. We are also the first to show how the popular thought of orthogonal projection can be realized in pure neuronal systems. Since our HLOP method is fully composed of neural circuits with neuronal operations and is effective for SNNs as well as more biologically plausible error propagation methods, it may be applied to neuromorphic computing systems. And the possible parallelism of the lateral connections and Hebbian learning with the forward and error propagation may better leverage its advantages. Although we mainly focus on gradient-based supervised learning for SNNs in this paper, HLOP may also be extended to other learning methods based on presynaptic activity traces. We expect that our approach can pave solid paths for building continual neuromorphic computing systems.

ACKNOWLEDGMENTS

Z. Lin was supported by National Key R&D Program of China (2022ZD0160300), the NSF China (No. 62276004), and the major key project of PCL, China (No. PCL2021A12). D. He was supported by National Science Foundation of China (NSFC62376007).

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

# A    DETAILS OF SNN TRAINING METHODS AND COMBINATION WITH HLOP

In this section, we provide a detailed introduction to SNN training methods used in our work and the combination of the proposed HLOP with them.

## A.1    DSR

DSR (Meng et al., 2022) is a training method based on the explicit encoding of spike trains and the analytical transformations between spike representations. For the LIF neuron model, DSR considers the discrete update equations as:

$$\begin{cases} u[t] = e^{-\frac{\Delta t}{\tau}} v[t-1] + \left(1 - e^{-\frac{\Delta t}{\tau}}\right) I[t], \\ s[t] = H(u[t] - V_{th}), \\ v[t] = u[t] - V_{th} s[t], \end{cases} \tag{2}$$

where $u[t]$ is the membrane potential before spiking, $s[t]$ is the output spike, $v[t]$ is membrane potential after spiking and resetting, $I[t]$ is the input current, $t$ is the time step index, $H$ is the Heaviside step function, $V_{th}$ is the threshold, $\Delta t$ is the discrete step, and $\tau$ is the time constant.

DSR defines the spike representation of the spike train with $T$ time steps as the weighted firing rate:

$$a[T] = \frac{V_{th} \sum_{t=1}^{T} \lambda^{T-t} s[t]}{\sum_{t=1}^{T} \lambda^{T-t} \Delta t}, \tag{3}$$

where $\lambda = e^{-\frac{\Delta t}{\tau}}$. For multi-layer feedforward neural networks, let $\mathbf{a}^l[T]$ denote the spike representations of neurons at layer $l$, DSR derives that $\mathbf{a}^l[T]$ approximates sub-differentiable mappings $\mathbf{z}^l$ with the analytical closed-form transformation as:

$$\mathbf{z}^l[T] = \text{clamp}\left(\frac{1}{\tau l} \mathbf{W}^{l-1} \mathbf{z}^{l-1}[T], 0, \frac{V_{th}^l}{\Delta t}\right), \tag{4}$$

where $\mathbf{W}^{l-1}$ is the weight matrix between layer $l-1$ and layer $l$. Then $\mathbf{a}^l[T]$ can be viewed as following such transformation as well.

With the explicit encoding $\mathbf{a}^l[T]$ of spike trains and the approximate closed-form transformations between $\mathbf{a}^l[T]$, DSR backpropagates errors and calculates gradients based on $\mathbf{a}^l[T]$ and their transformations, as shown in Fig. 6(b). The gradients for weights are calculated by $\frac{\partial L}{\partial \mathbf{W}^l} = \frac{\partial L}{\partial \mathbf{a}^N[T]} \prod_{j=0}^{N-l-2} \frac{\partial \mathbf{a}^{N-j}[T]}{\partial \mathbf{a}^{N-j-1}[T]} \frac{\partial \mathbf{a}^{l+1}[T]}{\partial \mathbf{W}^l}$, and can be rewritten in the form as $\nabla_{\mathbf{W}^l} L = \boldsymbol{\delta}^{l+1} \mathbf{a}^l[T]^\top$, where $\boldsymbol{\delta}^{l+1}$ is the backpropagated error signal. For static images, the inputs to SNNs are set as real-valued pixel values at each time step, which can be viewed as the input currents, and the loss is calculated based on the encoding $\mathbf{a}^N[T]$ at the last layer. To reduce the representation error, DSR additionally proposes to train the spike threshold and introduce a hyperparameter $\alpha$ to adjust the threshold and reset potential. We maintain these techniques following the default setting in the released code. Hyperparameters are set as the default value (Meng et al., 2022): $T = 20, V_{th} = 0.3, \tau = 1.0, \Delta t = 0.05, \alpha = 0.3, V_{th}^{bound} = 0.0005$, where $V_{th}^{bound}$ is a hyperparameter to bound the trainable threshold.

When applying our proposed HLOP to the DSR method, $\mathbf{a}^l[T]$ can be viewed as the activity traces and we act on them, as shown in Fig. 6(b). Note that $\mathbf{a}^l[T]$ is calculated based on spikes at each time step with different coefficients, so it is equivalent to applying HLOP to all spike signals: the calculation of the projection for weight update is the same. Considering Hebbian learning of lateral weights, there could be two choices to update weights: using the weighted firing rate (spike representation) for one calculation or using all spike trains at each time step for multiple calculations (i.e., Hebbian learning is carried out at each time step for multiple times). We verified both methods and their results are similar, as shown in Table 3. So we leverage the weighted firing rate by default for efficiency.

The thought to train SNNs with analytical transformations for the explicit encoding of spike trains is also shared by many works (Lee et al., 2016; Zhang & Li, 2019; Wu et al., 2021; Xiao et al., 2021; 2023; Mostafa, 2017; Comsa et al., 2020; Zhou et al., 2021). These works leverage (weighted) firing

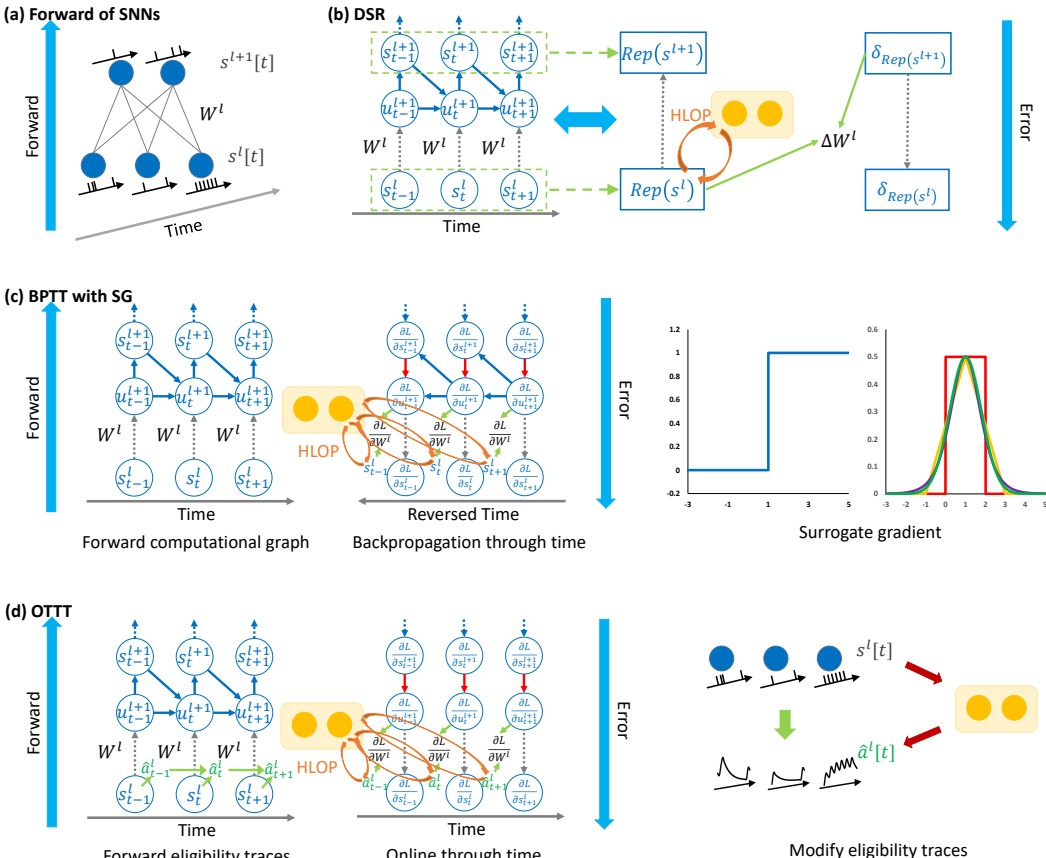

Figure 6: Illustration of applying HLOP to different supervised SNN training methods. (a) Illustration of spiking neural networks with temporal spike trains. (b) Training methods based on explicit coding schemes, e.g., DSR (Meng et al., 2022). Gradients are calculated by analytical transformations between spike representation ($Rep(s^l)$ in the figure), such as weighted firing rate. Activity traces correspond to the spike representation and HLOP can act on the presynaptic spike representation. (c) Training methods with backpropagation through time and surrogate gradients. Gradients are recursively calculated for all previous time steps. Activity traces correspond to spikes at each time step and HLOP can act on all presynaptic spike signals. (d) Training methods with online calculation through time and eligibility traces (Bellec et al., 2020; Xiao et al., 2022). Gradients are calculated with tracked traces and instantaneous errors. Activity traces correspond to the tracked traces and HLOP can act on all the traces. This only requires online modifying traces by lateral signals, i.e., neuronal traces will consider both spike responses for feedforward inputs and responses for lateral circuits.

Table 3: Results of average accuracy (ACC) and average backward transfer (BWT) with different Hebbian learning settings for the DSR SNN training method.

| Hebbian Learning Setting | PMNIST | | 10-split CIFAR-100 | |
|---|---|---|---|---|
| | ACC (%) | BWT (%) | ACC (%) | BWT (%) |
| weighted firing rate | 95.15 | −1.30 | 78.58 | −0.26 |
| spike trains | 95.23 | −1.34 | 78.42 | −0.41 |

rate / accumulated response for transformations (Lee et al., 2016; Zhang & Li, 2019; Zhou et al., 2021; Wu et al., 2021; Xiao et al., 2021; 2023) or consider the first spiking time as the encoding and derive the analytical transformations (Mostafa, 2017; Comsa et al., 2020; Zhou et al., 2021).

By treating the explicit encoding as activity traces, our proposed HLOP can be applied to all these methods similarly to DSR. HLOP provides a general way to support continual learning for training methods based on presynaptic traces, which is orthogonal to specific SNN training approaches.

## A.2 BPTT with SG

Backpropagation through time (BPTT) with surrogate gradients (SG) is a popular way to train SNNs without specific coding schemes (Shrestha & Orchard, 2018; Bellec et al., 2018; Wu et al., 2018; Zheng et al., 2021; Li et al., 2021; Fang et al., 2021; Yin et al., 2021). BPTT unfolds the discrete computational graph of SNNs through time and treats it similarly to recurrent neural networks, as shown in Fig. 6(c). The non-differentiable problem of the spiking function is tackled by leveraging the derivative of a smoothed function as a surrogate. Specifically, for the computation through time:

$$\begin{cases} \mathbf{u}^l\,[t+1] = \lambda(\mathbf{u}^l[t] - V_{th}\mathbf{s}^l[t]) + \mathbf{W}^{l-1}\mathbf{s}^{l-1}[t] + \mathbf{b}^l, \\ \mathbf{s}^l[t+1] = H(\mathbf{u}^l\,[t+1] - V_{th}), \end{cases} \tag{5}$$

the gradients are calculated as:

$$\frac{\partial L}{\partial \mathbf{W}^l} = \sum_{t=1}^{T} \frac{\partial L}{\partial \mathbf{s}^{l+1}[t]} \frac{\partial \mathbf{s}^{l+1}[t]}{\partial \mathbf{u}^{l+1}[t]} \left( \frac{\partial \mathbf{u}^{l+1}[t]}{\partial \mathbf{W}^l} + \sum_{\tau < t} \prod_{i=1}^{t-\tau} \left( \frac{\partial \mathbf{u}^{l+1}[t-i+1]}{\partial \mathbf{u}^{l+1}[t-i]} + \right. \right.$$
$$\left. \left. \frac{\partial \mathbf{u}^{l+1}[t-i+1]}{\partial \mathbf{s}^{l+1}[t-i]} \frac{\partial \mathbf{s}^{l+1}[t-i]}{\partial \mathbf{u}^{l+1}[t-i]} \right) \frac{\partial \mathbf{u}^{l+1}[\tau]}{\partial \mathbf{W}^l} \right), \tag{6}$$

where $\frac{\partial L}{\partial \mathbf{s}^{l+1}[t]}$ is the error backpropagated to layer $l+1$ at time $t$, and the non-differentiable terms $\frac{\partial \mathbf{s}^l[t]}{\partial \mathbf{u}^l[t]}$ will be replaced by surrogate derivatives, e.g., derivatives of rectangular or sigmoid functions (Wu et al., 2018): $\frac{\partial s}{\partial u} = \frac{1}{a_1}\text{sign}\left(|u - V_{th}| < \frac{a_1}{2}\right)$ or $\frac{\partial s}{\partial u} = \frac{1}{a_2}\frac{e^{(V_{th}-u)/a_2}}{(1+e^{(V_{th}-u)/a_2})^2}$ ($a_1$ and $a_2$ are hyperparameters). We leverage the sigmoid-like functions in our experiments and set $a_2 = 0.25$. The gradients can be rewritten in the form as $\nabla_{\mathbf{W}^l} L = \sum_{t=1}^{T} \hat{\boldsymbol{\delta}}^{l+1}[t]\hat{\mathbf{s}}^l[t]^\top$, where $\hat{\boldsymbol{\delta}}^{l+1}[t]$ is the error for each time step recursively backpropagated through time. For static images, the inputs to SNNs are also set as real-valued pixel values at each time step as the input currents, and the loss is calculated based on the firing rate at the last layer.

When applying our proposed HLOP to the BPTT with SG method, the activity traces are all spikes $\hat{\mathbf{s}}^l[t]$ at each time step, and HLOP acts on all spike signals, as shown in Fig. 6(c). Both projection and Hebbian learning are based on spikes, and the traces of spikes are all projected to formulate the projected gradient of $\nabla_{\mathbf{W}^l} L$.

## A.3 OTTT

OTTT (Xiao et al., 2022) is a training method to improve BPTT for temporally online learning. BPTT requires storing the computational graph unfolded over time and backpropagating through previous time steps, which is inconsistent with biological online learning and learning rules on neuromorphic hardware. Some works introduce eligibility traces to improve BPTT for online training through time (Zenke & Ganguli, 2018; Bellec et al., 2020; Bohnstingl et al., 2022; Xiao et al., 2022), in which OTTT is an efficient and scalable approach with more theoretical guarantees. Specifically, OTTT analyzes the gradients of BPTT and proposes to decouple the temporal dependency by tracking the presynaptic activities and leveraging instantaneous gradients as:

$$\hat{\mathbf{a}}^l[t] = \sum_{\tau \le t} \lambda^{t-\tau}\mathbf{s}^l[\tau], \quad \hat{\mathbf{a}}^l[t+1] = \lambda\hat{\mathbf{a}}^l[t] + \mathbf{s}^l[t+1], \tag{7}$$

$$\mathbf{g}_{\mathbf{u}^{l+1}}[t] = \left( \frac{\partial L[t]}{\partial \mathbf{s}^N[t]} \prod_{i=0}^{N-l-2} \frac{\partial \mathbf{s}^{N-i}[t]}{\partial \mathbf{s}^{N-i-1}[t]} \frac{\partial \mathbf{s}^{l+1}[t]}{\partial \mathbf{u}^{l+1}[t]} \right)^\top, \tag{8}$$

$$\nabla_{\mathbf{W}^l} L = \sum_{t=1}^{T} \mathbf{g}_{\mathbf{u}^{l+1}}[t]\hat{\mathbf{a}}^l[t]^\top, \tag{9}$$

where $L[t] = \frac{1}{T}\mathcal{L}\left(\mathbf{s}^L[t], \mathbf{y}\right)$ is the instantaneous loss and the total loss $L = \sum_{t=1}^{T} L[t]$ is the upper bound of the commonly used firing-rate-based loss when $\mathcal{L}$ is a convex function such as cross-entropy. With such calculation, OTTT does not require backpropagation through time and enables forward-in-time learning, as shown in Fig. 6(d).

When applying our proposed HLOP to these eligibility traces-based methods, the activity traces correspond to the tracked traces at each time step, and HLOP acts on them, as shown in Fig. 6(d). Note that the tracked traces are calculated based on spike trains, so the projection of traces only requires modifying eligibility traces by lateral signals, i.e., the neuronal traces consider both spike responses for feedforward inputs and responses for lateral circuits. Specifically, the traces are originally accumulated by $\hat{\mathbf{a}}^l[t+1] = \lambda\hat{\mathbf{a}}^l[t] + \mathbf{s}^l[t+1]$ for each neuronal spike, but will change to $\hat{\mathbf{a}}^l[t+1] = \lambda\hat{\mathbf{a}}^l[t] + \mathbf{s}^l[t+1] + \mathbf{s}^{l^-}[t+1]$ when HLOP is applied, considering the response $\mathbf{s}^{l^-}[t+1]$ for lateral circuits with the input signal $\mathbf{s}^l[t+1]$, e.g., $\mathbf{s}^{l^-}[t+1] = -\mathbf{H}_l^\top\mathbf{H}_l\mathbf{s}^l[t+1]$. In this way, both projection and Hebbian learning are based on spikes as well. The online training methods are more friendly for neuromorphic online learning, and the easy application of our HLOP with purely neuronal operations to these methods can pave a solid path for continual on-chip training of neuromorphic computing systems.

# B   DETAILS OF BACKPROPAGATION, FEEDBACK ALIGNMENT, AND SIGN SYMMETRIC

In this section, we provide a detailed introduction to backpropagation (BP), feedback alignment (FA), and sign symmetric (SS) used in our work.

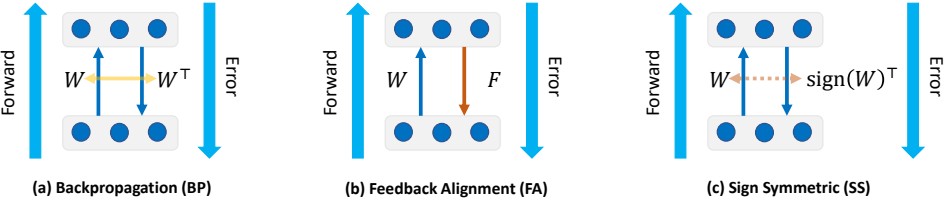

Figure 7: Illustration of different error propagation methods.

As shown in Fig. 7, BP, FA, and SS mainly differ in the way they back-propagate error signals across layers. BP follows the chain rule of gradient calculation to propagate errors. Specifically, for the feedforward calculation $\mathbf{y} = \mathbf{W}\mathbf{x}$, the error will be backpropagated by $\nabla_{\mathbf{x}}L = \mathbf{W}^\top\nabla_{\mathbf{y}}L$. This requires symmetric weights for forward and backward connections between neurons.

FA (Lillicrap et al., 2016) relaxes the symmetric weight requirement and proposes to leverage random feedback weights $\mathbf{F}$ to propagate error. They observe that training with random feedback weights can lead to the alignment of forward weights and feedback weights, enabling effective learning. FA replaces the calculation of $\nabla_{\mathbf{x}}L = \mathbf{W}^\top\nabla_{\mathbf{y}}L$ for each layer by $\nabla_{\mathbf{x}}L = \mathbf{F}\nabla_{\mathbf{y}}L$. There are other variants of FA such as direct feedback alignment (DFA) (Nøkland, 2016) etc., and we consider FA in experiments. The feedback weights $\mathbf{F}$ are initialized following the same distribution as the forward weights, e.g., the Kaiming uniform distribution.

SS (Xiao et al., 2018) also relaxes the strict symmetric weight requirement but does not use fully random feedback weights. They consider that it is possible to pass a little information about the sign direction between forward and feedback weights, and this enables SS to scale to large-scale models and tasks. SS replaces the calculation of $\nabla_{\mathbf{x}}L = \mathbf{W}^\top\nabla_{\mathbf{y}}L$ for each layer by $\nabla_{\mathbf{x}}L = s \times \text{sign}(\mathbf{W})^\top\nabla_{\mathbf{y}}L$, where $s$ is a scale factor to prevent gradient explosion. We follow the same setting in Xiao et al. (2018).

## C More Details of HLOP

Our HLOP method adds recurrent lateral connections for each layer of neural network models. For the $l$-th layer of SNNs, the output spike trains $\mathbf{s}^l[t]$ of feedforward neurons will also be passed to the lateral subspace neurons with skew-symmetric connection weights $\mathbf{H}_l$ and $-\mathbf{H}_l^\top$. For DSR, the spike trains are mainly treated as the specified spike representation for the calculation of HLOP, and for BPTT with SG and OTTT, spikes or eligibility traces at each time step are considered for HLOP. In the following, we first denote activity traces as $\mathbf{x}_l$ for simplicity.

When learning the first task, there is no consolidated subspace but only newly expanded subspace neurons with weights $\mathbf{H}_l'$. So there will be no projection for the feedforward synaptic update and it is still based on the activity trace $\mathbf{x}_l$. The lateral synaptic weight is updated by Hebbian learning as $\Delta\mathbf{H}_l' = \mathbf{y}_l'\mathbf{x}_l^\top + \mathbf{y}_l'\tilde{\mathbf{x}}_l^\top$, where $\mathbf{y}_l' = \mathbf{H}_l'\mathbf{x}_l, \tilde{\mathbf{x}}_l = -\mathbf{H}_l'^\top\mathbf{y}_l'$. Since the Hebbian learning can be viewed as stochastic gradient descent for minimizing an explicit objective function $\min_{\mathbf{H}_l'} \mathbb{E}_{\mathbf{x}_l} \left\| \mathbf{x}_l - \mathbf{H}_l'^\top\mathbf{H}_l'\mathbf{x}_l \right\|^2$ (Pehlevan et al., 2015), inspired by the momentum technique commonly used in gradient-based methods, we also introduce momentum update in practice. We use momentum to track the synaptic update and modify weights based on the momentum. The momentum is taken as 0.9 in practice. After learning each task, the newly expanded subspace neurons will be consolidated.

When learning successive tasks, consolidated subspace neurons with weights $\mathbf{H}_l$ will project the activity traces, and new subspace neurons with weights $\mathbf{H}_l'$ will be constructed with Hebbian learning. Let $\mathbf{y}_l = \mathbf{H}_l\mathbf{x}_l, \mathbf{x}_l^- = -\mathbf{H}_l^\top\mathbf{y}_l, \mathbf{y}_l' = \mathbf{H}_l'\mathbf{x}_l$, and $\mathbf{x}_l^{-\prime} = -\mathbf{H}_l'^\top\mathbf{y}_l'$ denote signals of recurrent lateral connections of consolidated and new subspaces, respectively. $\hat{\mathbf{x}}_l = \mathbf{x}_l + \mathbf{x}_l^-$ is the modified activity trace and the feedforward synaptic update is calculated based on the projected activity trace $\hat{\mathbf{x}}_l$, instead of the original one, as well as the error signal. $\tilde{\mathbf{x}}_l = \mathbf{x}_l^- + \mathbf{x}_l^{-\prime}$ is the integrated postsynaptic response to the recurrent connections for Hebbian update. Only the new lateral synaptic weight $\mathbf{H}_l'$ is updated by Hebbian learning as $\Delta\mathbf{H}_l' = \mathbf{y}_l'\mathbf{x}_l^\top + \mathbf{y}_l'\tilde{\mathbf{x}}_l^\top$, which can be viewed as minimizing the objective function $\min_{\mathbf{H}_l'} \mathbb{E}_{\mathbf{x}_l} \left\| \mathbf{x}_l - \mathbf{H}_l^\top\mathbf{H}_l\mathbf{x}_l - \mathbf{H}_l'^\top\mathbf{H}_l'\mathbf{x}_l \right\|^2$ to extract the new principal subspace not included in the existing subspace neurons. The momentum technique is also applied. And in practice, for each mini-batch data, we update lateral synaptic weights by $K$ times to accelerate convergence. We take $K = 5$ and the learning rate as 0.01 for Hebbian learning in experiments. After learning the new task, the new subspace neurons will be consolidated and the connection weights $\mathbf{H}_l'$ are concatenated into $\mathbf{H}_l$.

For convolutional operations, the shared kernel will linearly act on multiple patches of feature maps. We view the dimension of presynaptic inputs of feedforward neural networks as the dimension of these patches and view them as different input samples. Hebbian learning is similarly applied to the shared lateral connections between these patches and subspace neurons.

## D More Details of HLOP with Lateral Spiking Neurons

In this section, we provide more details about HLOP with lateral spiking neurons. We consider that the subspace neurons in the lateral circuits can generate high-frequency bursts in a short time and then the neuron model is approximated by the non-leaky integrate and fire (IF) model. The computation of the IF neuron model with input current $I$ is described as:

$$\begin{cases} u[t+1] = (u[t] - V_{th}s[t]) + I, \\ s[t+1] = H(u[t+1] - V_{th}), \end{cases} \tag{10}$$

It is easy to prove that the scaled output spiking rate of the IF model, i.e., $r[T] = \frac{V_{th}}{T}\sum_{t=1}^{T} s^l[t]$, is a quantization of the input when $I$ is positive, i.e., $r[T] = \frac{V_{th}}{T}\lfloor\frac{TI}{V_{th}}\rfloor$ with a clipping to the range $[0, V_{th}]$. If we initialize the membrane potential as 0 and modify the spiking threshold and reset potential as $\frac{V_{th}}{2}$ and $-\frac{V_{th}}{2}$, respectively, we can obtain a more precise quantization as $r[T] = \frac{V_{th}}{T}[\frac{TI}{V_{th}}]$. By considering the rate of high-frequency bursts and the response to the burst, we can obtain HLOP with lateral spiking neurons by taking the output $\mathbf{y}$ of subspace neurons as the quantization of $\mathbf{Hx}$ and the (scaled) response to burst spikes as $\mathbf{x}^- = -\mathbf{H}^\top\mathbf{y}$, as shown in Fig. 8(a).

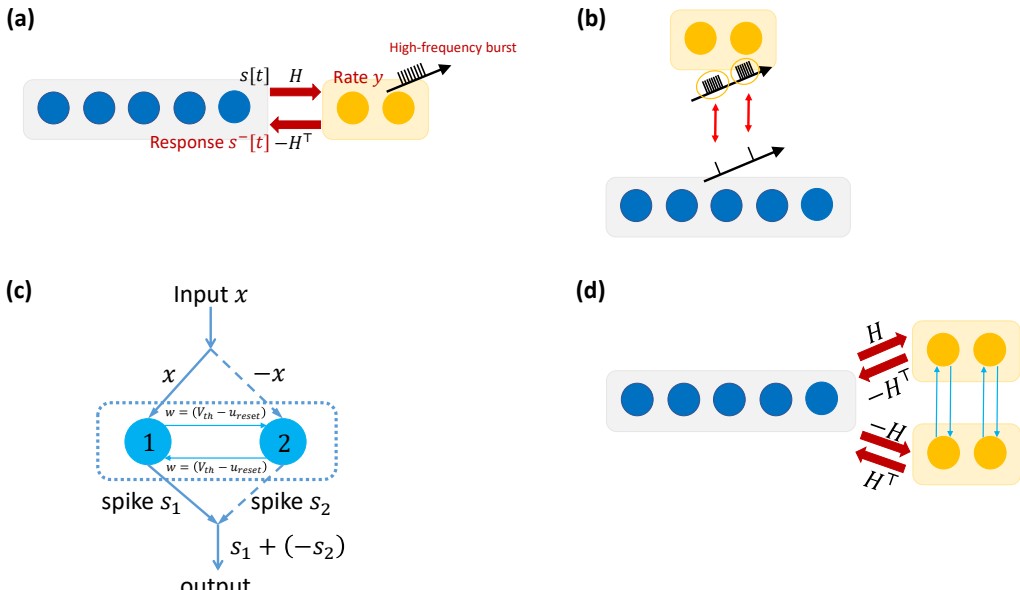

Figure 8: Illustration of HLOP with lateral spiking neurons. (a) HLOP with rate coding of high-frequency bursts of spiking subspace neurons. (b) Illustration of high-frequency bursts with a different time scale than feedforward spiking neurons. Each feedforward spike can correspond to a burst of lateral neurons. (c) Conceptual illustration of ternary neuron couples with common neuron models for ternary outputs. (d) Illustration of realizing ternary neuron couples for HLOP with opposite connection weights.

One problem is that the common neuron models only spike for positive inputs, limiting the ability to deal with negative signals. We require ternary spiking in order to imitate the original linear neurons of HLOP, i.e., the spiking function should be:

$$s[t+1] = T\left(u[t+1], V_{th}\right) = \begin{cases} 1, & u[t+1] > V_{th} \\ 0, & |u[t+1]| \leq V_{th} \\ -1, & u[t+1] < -V_{th} \end{cases}. \tag{11}$$

One possible approach is to enable such ternary spiking on hardware. Another way is to realize the ternary output by neuron couples with the common positively spiking neuron model, as introduced in (Xiao et al., 2023). The conceptual illustration of such ternary neuron couples is shown in Fig. 8(c), where two neurons receive opposite inputs and share a small recurrent connection to synchronize resetting. The operation of taking negative can be realized by reconnecting neurons with opposite weights, i.e., the two coupled neurons are oppositely connected to other neurons. Fig. 8(d) illustrates such realization for HLOP modules. This may require weight sharing between connections to neuron couples.

With such a design, we can leverage spiking neurons to imitate linear neurons except for some quantization. We simulate our spiking HLOP in experiments by quantizing the output of subspace neurons in experiments. Specifically, the output $y$ is quantized as $\hat{y} = scale \times \frac{\left\lfloor \frac{\text{clamp}(y, -scale, scale)}{scale} \times T_l \right\rfloor}{T_l}$. Both projection and Hebbian learning are based on $\hat{y}$. Since it is expected to be realized by high-frequency bursts in a short time, the spiking HLOP can have a different time scale than spiking neurons in the feedforward network as shown in Fig. 8(b), and projection and Hebbian learning with rate coding of burst spikes can be carried out for all spikes/eligibility traces in the feedforward network at each discrete time step which has a larger temporal interval.

# E  EXPERIMENTAL DETAILS

Our experiments are conducted on PMNIST, 10-split CIFAR-100, 5-Datasets, and 20-split miniImageNet. The PMNIST dataset is a variant of the MNIST dataset and each task is a different random

permutation of the original pixels. The 10 sequential tasks share the same classifier with 10 classes' output. This single-head classifier setting will require orthogonal projection for the update of the classifier. The 10-split CIFAR-100 dataset splits 100 classes of the CIFAR-100 dataset into 10 tasks with 10 classes per task. 5-Datasets is a sequential of CIFAR-10, MNIST, SVHN, Fashion MNIST, and notMNIST, where each task has 10 classes. The 20-split miniImageNet dataset splits 100 classes of the miniImageNet dataset into 20 tasks with 5 classes per task. These three settings use multi-head classifiers for different tasks and do not need projection for classifiers. We normalize the inputs based on the dataset statistics and do not use any data augmentation. We do not split the training and validation set and take the model at the last training epoch as the final model for testing because it would be vague to determine the best model considering both the accuracy and fitness of Hebbian learning.

As for the network structures, for experiments on PMNIST, we consider the fully connected networks with two hidden layers 784-800-800-10, and all tasks share the same classifier. For experiments on 10-split CIFAR-100, we consider three convolutional hidden layers whose kernel size is 3 and whose channels are 64, 128, and 256, and different tasks have different final classifiers. A batch normalization (BN) is added after each convolutional operation, which follows the practice in DSR (Meng et al., 2022) and can be absorbed into linear layers after training. Average pooling is added after each spiking layer. For experiments on 5-Datasets, we consider a reduced ResNet-18 network following previous work (Saha et al., 2021), whose base channel is set as 20, and different tasks have different final classifiers. For experiments on 20-split miniImageNet, we also consider the reduced ResNet-18 network, and the first convolution has stride 2 and kernel size 5, similar to Saha et al. (2021). For SNN models, the input and output encodings follow the practice in different training methods (Meng et al., 2022; Xiao et al., 2022) when we use them (BPTT with SG and OTTT follow the same setting as in Xiao et al. (2022) and BPTT with SG uses the rate-based loss), and other details and hyperparameters also follow them, e.g., we take the default 20 time steps for DSR while the default 6 time steps for BPTT with SG and OTTT.

For the continual learning setting, we only track the statistics of BN for the first task and will fix them for the successive tasks following the practice in Mallya & Lazebnik (2018), in order to avoid interference with old tasks. The learnable threshold in the DSR method is also learned in the first task and fixed in the following tasks. We follow the common practice to track statistics during training and use stored statistics during testing.

Our comparison methods include baseline, small memory replay (MR), EWC (Kirkpatrick et al., 2017), HAT (Serra et al., 2018), and GPM (Saha et al., 2021). The baseline method simply trains models for each task sequentially. The MR method randomly stores 50 input samples for each class in current tasks and replays all stored samples for learning with additional 20 epochs after each new task (for miniImageNet, the stored sample size is 5 for each class, corresponding to one percent of data similar to other settings). EWC and HAT are implemented by adapting the official code in Serra et al. (2018), and GPM is implemented by adapting the official implementation. We also consider the Multitask method as an upper bound that simultaneously learns all tasks at once. All methods share the same training setting.

For PMNIST, we consider the online setting and train models for 1 epoch by SGD with a batch size of 64 and a learning rate of 0.1. The Multitask setting for PMNIST is slightly different as the batch size of each task is set as 8 to enable a similar number of weight updates under the online setting and the learning rate is taken as 0.01. The hyperparameters for HAT, EWC, and GPM are taken as the default in the code. For HLOP, both hidden layers and the classifier require orthogonal projection, and the number of subspace neurons for the three layers is set as [80, 200, 100] (from bottom to top) at the first task. For each later task, the number of newly expanded subspace neurons is initially [70, 70, 70] for each layer and is reduced by 20 every 3 tasks.

For 10-split CIFAR-100, we train all models for 200 epochs by SGD with momentum 0.9. The batch size is 64, and the learning rate is 0.01 with the cosine annealing scheduler to 0 as well as a linear warm-up in the first 5 epochs. The hyperparameters for EWC, HAT, and GPM are taken as the default in the code. For HLOP, the number of subspace neurons is set as [6, 100, 200] for each layer at the first task (around 1/5 as the dimension of presynaptic inputs), and the number of newly expanded subspace neurons for successive tasks is set as [2, 20, 40]. For the combination between HLOP and MR, we do not update lateral connections during replay.

For 5-Datasets, we train all models for 100 epochs by SGD with momentum 0.9. The batch size is 64, and the initial learning rate is 0.1 for the first task and 0.01 for successive tasks, with the cosine annealing scheduler to 0. The hyperparameters for EWC, HAT, and GPM are taken as the default in the code. For HLOP, the number of subspace neurons is set as [6, [40, 40], [40, 40], [40, 100, 6], [100, 100], [100, 200, 8], [200, 200], [200, 200, 16], [200, 200]] for each presynaptic layer in ResNet-18 at the first task (around 1/5 as the dimension of presynaptic inputs) and the number of newly expanded neurons is the same.

For 20-split miniImageNet, we train all models for 100 epochs by SGD with momentum 0.9. The batch size is 64, and the initial learning rate is 0.1 for the first task and 0.01 for successive tasks, with the cosine annealing scheduler to 0. The learning rate for multitask is 0.01. The hyperparameters for EWC, HAT, and GPM are taken as the default in the code/paper. For HLOP, the number of subspace neurons is set as [24, [90, 90], [90, 90], [90, 180, 10], [180, 180], [180, 360, 20], [360, 360], [360, 720, 40], [720, 720]] for each presynaptic layer in ResNet-18 at the first task (around 1/2 as the dimension of presynaptic inputs) and the number of newly expanded neurons is [2, [6, 6], [6, 6], [6, 12, 1], [12, 12], [12, 24, 2], [24, 24], [24, 48, 4], [48, 48]] (around 1/30) for the first five successive tasks and [0, [2, 2], [2, 2], [2, 4, 0], [4, 4], [4, 8, 0], [8, 8], [8, 16, 0], [16, 16]] (around 1/90) for others. The number is decreasing because there are many shared subspaces between different tasks (miniImageNet has similar input domains) and the considered network capacity is not large enough.

Currently, the number of subspace neurons is a hyperparameter for HLOP, which is similar to the threshold hyperparameter in GPM (Saha et al., 2021). It may be determined with some prior knowledge about the task difficulty, e.g., first train the network for a short time to decide the roughly required number (for example, the ratio of the dimension of presynaptic inputs) and then formally perform training with Hebbian learning. It can be interesting future work to study how to automatically and adaptively determine the number with neuronal mechanisms.

All code implementations are based on the PyTorch framework and experiments are carried out on one NVIDIA GeForce RTX 3090 GPU. All experiments and dataset split are under the same random seed 2022.

# F  MORE DISCUSSION

## F.1  DISCUSSION ON HEBBIAN LEARNING

One major component of our method is Hebbian learning. Hebbian learning is often related to memory modeling, e.g., Hopfield network (Hopfield, 1982) with Hebbian-type learning can serve as associative memory systems. In this work, we provide a new perspective on how Hebbian learning can support advanced learning abilities, which is not in an associative approach. Extraction of the principal subspace of streaming data is to some extent similar to memory but is not in an associative approach, and can well fit into our projection-based continual learning method. This work mainly focuses on Hebbian learning based on neuronal spiking activities, which can be shown to optimize synaptic weights for principal component analysis (Oja, 1982; 1989). In biological systems, spike-timing-dependent plasticity (STDP) is the often observed Hebbian learning considering the time of spikes (Caporale et al., 2008). However, STDP has not been shown to optimize an explicit objective. It is interesting future work to study if the biological STDP rule can be similarly leveraged to systematically promote advanced abilities such as continual learning.

A basic focus of our work is neuromorphic computing. Neuromorphic computing aims at the computation inspired by the structure and function of the human brain. At the hardware level, neuromorphic chips are designed to imitate biological neurons (Akopyan et al., 2015; Davies et al., 2018; Pei et al., 2019), such as their spiking property and synaptic connections with local storage of weights for in-memory computation, for highly energy-efficient and parallel event-driven computation with avoidance of frequent memory transpose (its computation architecture is expected to be different from the commonly used hardware with von Neumann architecture such as CPU or GPU). At the algorithm level, we are interested in developing methods compatible with some properties and operations of neurons so that they are possible for deployment on the hardware. Also, since neuromorphic hardware is under development considering software-hardware co-design, most algorithms are simulated on common hardware (e.g., GPU) while considering neuromorphic properties. From the perspective of neuromorphic computing, Hebbian learning can be mapped to the computation of

neurons and synapses and implemented on neuromorphic hardware, so it is more suitable for SNNs than other PCA approximation algorithms.

### F.2 DISCUSSION OF LIMITATIONS/CHALLENGES

As for our method, HLOP introduces additional computational costs for learning the lateral weights of each layer during training, which will increase training costs in our current implementation codes. While this process can theoretically be parallel to the normal forward-backward propagation of network training as discussed in Section 4.1, such parallelization may not be easily realized in established deep learning libraries such as PyTorch. So for our implementation of GPU training, the training time would be slightly longer and the evaluation of parallelization is lacking. It can be future work on low-level code optimization or consideration of asynchronously parallel neuromorphic computing hardware. Additionally, HLOP currently requires a manual specification for the number of subspace neurons, which may be improved to automatically and adaptively determine the allocation. For example, the Generalized Hebbian Algorithm can perform Gram-Schmidt orthonormalization on the rows of the weight matrix, which may help to sweep out unnecessary neurons and connections, but it requires some non-local information and may not be directly suitable to our method. It can be interesting for future work to study improvements.

As for the evaluation, similar to other gradient projection methods, this work mainly focuses on task-incremental and domain-incremental continual learning settings. There is another class-incremental setting which, as discussed in Section 2, inevitably requires some kind of replay of old experiences for good performance since it expects explicit comparison between new classes and old ones. So following previous works, our evaluation mainly focuses on task- and domain-incremental settings. It can be future work to study if HLOP can be combined with some replay methods or context-dependent processing modules similar to biological systems (e.g., with a task classifier) for better class-incremental tasks.

## G ADDITIONAL RESULTS

### G.1 DOMAIN-CL ON 5-DATASETS

To further show the scalability of our method for the domain-incremental setting on larger datasets and networks apart from PMNIST, we supplement the domain-CL results on 5-Datasets. Compared with the task-incremental settings, it shares the final classifier for all tasks, i.e., considering a single-head classifier. All models are trained for 10 epochs and MR also performs 10 epochs for additional replay training. As shown in Table 4 (where HAT requires task ID and is not feasible), HLOP significantly outperforms other methods, indicating the superiority of our method.

Table 4: Comparison results on the Domain-CL 5-Datasets.

| Method | ACC (%) | BWT (%) |
|---|---|---|
| *Multitask* | *82.81* | */* |
| Baseline | 32.46 | −74.86 |
| MR | 70.26 | −24.12 |
| EWC | 35.80 | −65.18 |
| HAT | N.A. | N.A. |
| GPM | 45.62 | −56.28 |
| **HLOP (ours)** | **79.59** | **−9.94** |

### G.2 COMPARISON BETWEEN ANNs AND SNNs

To further study whether the improvement of our method over others mainly comes from the better combination with SNNs or the continual learning ability, we supplement all the corresponding results of ANNs that replace the spiking neuron by the ReLU activation.

As shown in Table 5, the performance of some methods can differ considering ANNs and SNNs. Particularly, on 5-Datasets, HLOP has more improvements for SNNs. It may imply that our method

Table 5: Comparison results between ANNs and SNNs.

| Network | Method | PMNIST[1] | | 10-split CIFAR-100 | | miniImageNet | | 5-Datasets | |
|---|---|---|---|---|---|---|---|---|---|
| | | ACC | BWT | ACC | BWT | ACC | BWT | ACC | BWT |
| SNN | *Multitask*[2] | *96.15* | */* | *79.70* | */* | *79.05* | */* | *89.67* | */* |
| | Baseline | 70.61 | −28.88 | 75.13 | −4.33 | 40.56 | −32.42 | 45.12 | −60.12 |
| | MR | 92.53 | −4.73 | 77.67 | −1.01 | 58.15 | −8.71 | 79.66 | −14.61 |
| | EWC | 91.45 | −3.20 | 73.75 | −4.89 | 47.29 | −26.77 | 57.06 | −44.55 |
| | HAT | N.A. | N.A. | 73.67 | −0.13 | 50.11 | −7.63 | 72.72 | −22.90 |
| | GPM | 94.80 | −1.62 | 77.48 | −1.37 | 63.07 | −2.57 | 79.70 | −15.52 |
| | **HLOP (ours)** | 95.15 | −1.30 | 78.58 | −0.26 | 63.40 | −0.48 | 88.65 | −3.71 |
| ANN | *Multitask*[2] | *96.81* | */* | *80.33* | */* | *82.05* | */* | *88.65* | */* |
| | Baseline | 73.55 | −25.67 | 71.47 | −7.82 | 38.87 | −31.58 | 68.54 | −30.32 |
| | MR | 92.44 | −5.00 | 75.59 | −2.68 | 55.15 | −6.79 | 83.98 | −9.29 |
| | EWC | 90.16 | −3.46 | 72.49 | −6.01 | 47.41 | −21.46 | 68.60 | −29.90 |
| | HAT | N.A. | N.A. | 71.31 | 0.00 | 56.98 | −1.64 | 88.40 | −3.63 |
| | GPM | 94.92 | −1.56 | 77.56 | −1.42 | 65.13 | −0.96 | 84.88 | −9.17 |
| | **HLOP (ours)** | 95.25 | −1.28 | 78.98 | −0.34 | 66.80 | 1.93 | 88.68 | −2.74 |

[1] Experiments are under the online setting, i.e., each sample only appears once (except for the Memory Replay method), and the domain-incremental setting.
[2] Multitask does not adhere to the continual learning setting and can be viewed as an upper bound.

can be better combined with spikes, for example probably because subspaces expanded by spike signals are harder to learn and therefore our Hebbian learning performing streaming PCA has more advantages over GPM that only performs SVD/PCA on a small batch of data which can be biased. Also, our method is promising for ANNs as well. As there is no systematic study on the difference between the continual learning performance of ANNs and SNNs, it can be interesting for future work to study it.

