# OpenReview forum: "Hebbian Learning based Orthogonal Projection for Continual Learning of Spiking Neural Networks"
_ICLR.cc/2024/Conference — ICLR 2024 poster_

### Official Review · Reviewer_Me3Q · 2023-10-27

**Soundness:** 4 excellent
**Presentation:** 3 good
**Contribution:** 3 good
**Rating:** 8
**Confidence:** 4

**Summary:**

The paper applies Oja's rule to implement gradient projection for continual learning. The method uses lateral circuits to avoid storing gradient explicitly, and works with spiking networks.

**Strengths:**

To the best of my knowledge, the method is novel. Orthogonalized gradients are known to perform well for continual learning, and Oja's rule is known to do PCA, but (as far as I know) they've never been combined like in this paper. The resulting method doesn't need to explicitly store gradients, and computes the projection matrix on the go with bio plausible operations (and not recursive least squares). This is an advantage over other projection-based methods.

The method only uses Hebbian and anti-Hebbian plasticity, which makes it suitable for neuromorphic hardware and also biologically plausible. There's a caveat for biological networks though: the lateral circuits are only active during the backward pass, and don't interfere with the forward one. However, an exact biological implementation seems out of scope for this work.

Performance: on all (standard) benchmarks the method performs very well and usually outperforms other algorithms.

**Weaknesses:**

Some (not very critical) weaknesses:

1. The method needs to create new subspaces for each task, and then coordinate activity in a new subspace with the old ones. It's not clear if that can scale to many tasks (e.g. due to noise in PCA through Oja's rule)
2. The lateral connectivity doesn't influence forward propagation (and it shouldn't due to projections), which might make it hard to implement with real neurons (not so sure about neuromorphic chips).
3. There are no evaluations on non-spiking networks, so it's not clear if the performance improvement over other methods is due to those being poorly suited for spikes or this method being better at continual learning.

**Questions:**

The proposed architecture looks like a model of memory, since neurons only update their weights if they haven't seen a specific input before. Can all task-specific lateral circuits be combined into a single associative memory module, like a Hopfield net, with each new memory being a new $y$ for the task?

Before Sec. 2:
>  Our results indicate that lateral circuits, which are long ignored by popular feedforward neural network

I’d say that lateral circuits are often present implicitly through normalization layers.

Tab. 2: boldface should be used for the best performing method in a column, not the author's method.

---

> ### Author Response · Authors · 2023-11-19
> **Response to Reviewer Me3Q (Part 1/2)**
>
> Thank you for appreciating our work and providing valuable comments. We respond to your comments and questions as follows.
>
> 1. About scalability to many tasks.
>
> Our experiments verified effectiveness on common datasets with many tasks: PMNIST (10 tasks), CIFAR-100 (10 tasks), miniImageNet (20 tasks), 5-Datasets (5 tasks), also with similar or distinct input domains and online or multi-epoch training settings. So it can scale to many tasks.
>
> Actually, although there can be slight noise for PCA, our method enables unbiased streaming PCA that leverages all data, while previous gradient projection methods either only use a small batch of data to perform SVD and PCA [1,2] which can have larger noise and bias, or use recursive least square algorithm [3] that introduces a parameter $\alpha$ for calculation of inverses which will have bias too. Our method has improvements over them.
>
> 2. The lateral connectivity does not influence forward propagation.
>
> Yes, our lateral connections with subspace neurons mainly modify the activity traces (i.e., eligibility traces for SNNs under some training algorithms) of feedforward neurons for weight update, and do not influence the model’s forward propagation. While it mainly influences the calculation of gradients, this does not mean that it is only active during the backward pass – as discussed in Section 4.1, since HLOP only modifies presynaptic activity traces, this lateral projection can be parallel with the forward and backward propagation of the main network. That means, during the forward propagation of the main network, lateral connections can be simultaneously activated to modify traces of feedforward neurons, and once global error signals reach, weight updates are calculated based on the error signal and modified eligibility traces. We may make some conjectures that with some mechanisms such as the refractory period or adaptive threshold, the (fast) lateral connection may not induce spike signals for forward propagation but influence eligibility traces. It can be interesting future work to further consider if there can be biological correspondence and it is currently out of scope for this work.
>
> 3. About evaluations on non-spiking networks.
>
> Thank you for your valuable suggestion. Here we supplement all the corresponding results of ANNs that replace the spiking neuron by ReLU activation.
>
> | Neural Network | Method | PMNIST (ACC, BWT) | CIFAR-100 (ACC, BWT) | miniImageNet (ACC, BWT) | 5-Datasets (ACC, BWT) |
> | :----: | :----: | :----: | :----: | :----: | :----: |
> | SNN | *Multitask* | *96.15*, / | *79.70*, / | *79.05*, / | *89.67*, / |
> | SNN | Baseline | 70.61, -28.88 | 75.13, -4.33 | 40.56, -32.42 | 45.12, -60.12 |
> | SNN | MR | 92.53, -4.73 | 77.67, -1.01 | 58.15, -8.71 | 79.66, -14.61 |
> | SNN | EWC | 91.45, -3.20 | 73.75, -4.89 | 47.29, -26.77 | 57.06, -44.55 |
> | SNN | HAT | N.A., N.A. | 73.67, -0.13 | 50.11, -7.63 | 72.72, -22.90 |
> | SNN | GPM | 94.80, -1.62 | 77.48, -1.37 | 63.07, -2.57 | 79.70, -15.52 |
> | SNN | **HLOP (ours)** | 95.15, -1.30 | 78.58, -0.26 | 63.40, -0.48 | 88.65, -3.71 |
> | ANN | *Multitask* | *96.81*, / | *80.33*, / | *82.05*, / | *88.65*, / |
> | ANN | Baseline | 73.55, -25.67 | 71.47, -7.82 | 38.87, -31.58 | 68.54, -30.32 |
> | ANN | MR | 92.44, -5.00 | 75.59, -2.68 | 55.15, -6.79 | 83.98, -9.29 |
> | ANN | EWC | 90.16, -3.46 | 72.49, -6.01 | 47.41, -21.46 | 68.60, -29.90 |
> | ANN | HAT | N.A., N.A. | 71.31, 0.00 | 56.98, -1.64 | 88.40, -3.63 |
> | ANN | GPM | 94.92, -1.56 | 77.56, -1.42 | 65.13, -0.96 | 84.88, -9.17 |
> | ANN | **HLOP (ours)** | 95.25, -1.28 | 78.98, -0.34 | 66.80, 1.93 | 88.68, -2.74 |
>
> As shown in the results, the performance of some methods can differ considering ANNs and SNNs. Particularly, on 5-Datasets, HLOP has more improvements for SNNs. It may imply that our method can be better combined with spikes, for example probably because subspaces expanded by spike signals are harder to learn and therefore our Hebbian learning performing streaming PCA has more advantages over GPM that only performs SVD/PCA on a small batch of data which can be biased. Also, our method is promising for ANNs as well. As there is no systematic study on the difference between the continual learning performance of ANNs and SNNs, it can be interesting for future work to study it.
>
> 4. About associative memory module.
>
> As discussed in Appendix F, our Hebbian learning to extract principal subspaces of streaming data is to some extent similar to memory but is not in an associative approach, and can well fit into our projection formulation. Currently, we have no idea if an associative memory module can fit into the projection formulation, because in the task we mainly require some information about the general subspace for restriction rather than specific associative memory for a given input.

---

> > ### Author Response · Authors · 2023-11-19
> > **Response to Reviewer Me3Q (Part 2/2)**
> >
> > 5. Lateral circuits are often present implicitly through normalization layers.
> >
> > Thank you for pointing out this. Yes, some normalization layers such as Local Response Normalization can be viewed as an implicit lateral circuit. We delete the clause in the sentence for precision.
> >
> > 6. Boldface in Table 2.
> >
> > Thank you for your kind suggestion. We modify the boldface for the best result in this revision.
> >
> > [1] Gradient projection memory for continual learning. ICLR, 2021.
> >
> > [2] TRGP: Trust region gradient projection for continual learning. ICLR, 2022.
> >
> > [3] Continual learning of context-dependent processing in neural networks. Nature Machine Intelligence, 2019.

---

> > > ### Comment · Reviewer_Me3Q · 2023-11-21
> > >
> > > Thank you for the response! I think it addressed all of my concerns, so I'm happy with the current score of 8.
> > >
> > > I also read the concerns raised by other reviewers, but I believe they're mostly addressed by the paper/responses. I should also note that the new ANN experiments significantly improve the paper.

---

### Official Review · Reviewer_Txkh · 2023-10-29

**Soundness:** 3 good
**Presentation:** 3 good
**Contribution:** 3 good
**Rating:** 6
**Confidence:** 3

**Summary:**

This paper proposes Hebbian learning based orthogonal projection (HLOP) as a novel method for implementing orthogonal projection using neuronal operations. Building upon the method of calculating projection matrices through SVD proposed by Saha et al. (2021), HLOP combines the properties of Hebbian learning to approximate orthogonal projection using learned weight matrices. This is the first approach that fully utilizes neuronal operations for implementing orthogonal projection. Furthermore, HLOP outperforms several baseline methods on multiple continual learning datasets, demonstrating the reliability of the proposed method through experimental results.

**Strengths:**

1.The paper introduces for the first time a method that fully utilizes neuronal operations to approximate orthogonal projection matrices, and achieves the best performance surpassing multiple baseline methods on various datasets in continual learning task.
2.The implementation based on Hebbian learning can seamlessly integrate with different training methods and aligns well with the parallel local learning approach of neuromorphic chips.
3.HLOP can be directly applied to SNNs, providing a new approach for continual learning in SNNs.

**Weaknesses:**

1.The method lacks sufficient detail in the methodology section. Providing complete formulas or a specific illustrative example would enhance understanding.
2.The baselines compared in the study include EWC (Kirkpatrick et al., 2017), HAT (Serra et al., 2018), and GPM (Saha et al., 2021). It would be valuable to investigate if there are recent works that achieve better results than these methods.
3.Although the continual learning approach in this study relies solely on neuronal operations without directly utilizing past data, the increasing number of subspace neurons with each new task learned implies a form of data compression and storage to some extent.
4. Although the paper uses Hebbian learning to achieve a pure neuronal operation method, it seems that the newly added subspace neurons do not participate in the model's forward process. If this is the case, then despite being a pure neuronal operation approach, it is essentially just an estimation method for orthogonal projection matrices.
5. While the paper demonstrates the effectiveness of HLOP, it does not provide an explanation or analysis of why the weight matrices obtained through Hebbian learning can serve as a substitute for the orthogonal projection matrix.

**Questions:**

See weaknesses.

---

> ### Author Response · Authors · 2023-11-19
> **Response to Reviewer Txkh**
>
> Thank you for your valuable comments. We respond to your comments and questions as follows.
>
> 1. Method details.
>
> In Section 4, we mainly illustrate the procedure of our method in Figures 1 and 2, and formulas are in the texts due to the space limit. More detailed descriptions of HLOP can be found in Appendix C, and details of the combination with different SNN training methods are presented in Appendix A.
>
> 2. Baselines.
>
> Thank you for the suggestion. Since not many continual learning methods consider SNNs and we do not find SNN baselines on these common datasets, we mainly implement and compare representative methods of different kinds of methods (i.e., memory replay, regularization, and gradient projection) based on their released codes under our SNN settings, as we cannot exhaustingly reimplement all ANN continual learning methods. Note that the performance of some methods can differ considering ANNs and SNNs (please see our response to Reviewer Me3Q).
>
> There are some recent works [1] improving the gradient projection method GPM [2], which is our sota baseline. We try to test TRGP [1] based on their released code under our setting. On PMNIST and CIFAR-100, it does not outperform GPM as shown below, while on miniImageNet and 5-Datasets, we encounter failures that training fails after one or two tasks (i.e., the performance for new tasks is only random guess, but old tasks are maintained). It may probably be the difference between ANN and SNN, or hyperparameters and training settings, or some bugs in the code. Given the limited time, we cannot thoroughly analyze their method and codes. We think that current comparison is enough to show our superiority.
>
> | Method | PMNIST (ACC, BWT) | CIFAR-100 (ACC, BWT) |
> | :----: | :----: | :----: |
> | GPM | 94.80, -1.62 | 77.48, -1.37 |
> | TRGP | 94.56, -1.08 | 76.77, -2.61 |
> | **HLOP (ours)** | 95.15, -1.30 | 78.58, -0.26 |
>
> 3. Subspace neurons and data storage.
>
> Yes, our subspace neurons to some extent represent memory of some information of past tasks, and previous gradient projection method is also called Gradient Projection Memory [2], because we need information (of past tasks) to guide the restriction on weight update to solve forgetting. However, it is not explicit memory and thus does not require storing raw data which may bring concerns about data privacy [2].
>
> 4. Subspace neurons do not participate in the model’s forward process.
>
> Our lateral connections with subspace neurons mainly modify the activity traces (i.e., eligibility traces for SNNs under some training algorithms) of feedforward neurons for weight update, and do not influence the model’s forward propagation. While it mainly influences the calculation of gradients, this does not mean that it is only active during the backward pass – as discussed in Section 4.1, since HLOP only modifies presynaptic activity traces, this lateral projection can be parallel with the forward and backward propagation of the main network. That means, during the forward propagation of the main network, lateral connections can be simultaneously activated to modify traces of feedforward neurons, and once global error signals reach, weight updates are calculated based on the error signal and modified eligibility traces. We may make some conjectures that with some mechanisms such as the refractory period or adaptive threshold, the (fast) lateral connection may not induce spike signals for forward propagation but influence eligibility traces. It is interesting future work to further consider if there is biological correspondence and it is currently out of scope for this work.
>
> 5. Why weight matrices obtained through Hebbian learning can serve as a substitute for orthogonal projection matrix.
>
> As mentioned in Section 4.1, this is the known result that Hebbian learning can extract principal components of streaming inputs [3,4]. It has been shown that our considered formulation enables weights to converge to a dominant principal subspace [4]. In Appendix C, we also mentioned an intuitive explanation: Hebbian learning can be viewed as stochastic gradient descent for minimizing the objective function
>
> $ \min_{\mathbf{H}_l'} \mathbb{E} \left\lVert \mathbf{x}_l - \mathbf{H}_l^{\prime\top}\mathbf{H}_l'\mathbf{x}_l \right\rVert^2 $,
>
> and in our setting that updates new lateral weights with Hebbian learning, it can be viewed as minimizing
> $ \min_{\mathbf{H}_l'} \mathbb{E} \left\lVert \mathbf{x}_l - \mathbf{H}_l^\top\mathbf{H}_l\mathbf{x}_l - \mathbf{H}_l^{\prime \top}\mathbf{H}_l'\mathbf{x}_l \right\rVert^2 $
> to extract the new principal subspace not included in the existing subspace neurons.
>
> [1] TRGP: Trust region gradient projection for continual learning. ICLR, 2022.
>
> [2] Gradient projection memory for continual learning. ICLR, 2021.
>
> [3] Simplified neuron model as a principal component analyzer. Journal of Mathematical Biology, 1982.
>
> [4] Global analysis of oja’s flow for neural networks. IEEE Transactions on Neural Networks, 1994.

---

### Official Review · Reviewer_braR · 2023-10-30

**Soundness:** 3 good
**Presentation:** 3 good
**Contribution:** 3 good
**Rating:** 6
**Confidence:** 4

**Summary:**

This work proposes a task-incremental continual learning (CL) method for spiking neural networks.
It can be categorized as a CL approach based on orthogonal gradient projection.
In orthogonal gradient projection approaches, the update $\Delta \mathbf W^P$ of each layer is projected to a subspace to minimize changes to the outputs for previous tasks.
This is achieved by performing PCA on the input data for each layer and projecting the gradient to the subspace orthogonal to the principal subspace of the input data.

The main technical novelty of this work is the application of a Hebbian learning rule to perform PCA.
This rule is claimed to be more suitable for neuromorphic hardware.

**Strengths:**

- The paper is clearly written and easy to follow.
- The combination of spiking neural networks and continual learning seems interesting, although this is not the first work to address such problems.
- Code is provided in the supplementary material.

**Weaknesses:**

### Not Comparing Other Approximations of PCA

The essence of the proposed Hebbian approach is to perform PCA.
While there is a huge literature on more efficient approximate PCA with various forms of tradeoffs, the Hebbian approach is just one of such variants of PCA.

I think the authors need to justify why their Hebbian approach is particularly suitable for spiking neural networks.
They vaguely argue that the Hebbian rule only requires "neuronal operations," but the neuroscience-backed algorithm eventually boils down to some matrix-vector arithmetic, just like many other approximate PCA algorithms.
Currently, I do not see any reason that the Hebbian rule should be more suitable for spiking neural networks while others are not.

### Task-Incremental Settings

This paper exclusively focuses on the task-incremental settings.
Task-incremental CL is often considered the most naive and easiest form of CL.
Especially, providing task IDs even at test time is far from realistic and significantly reduces its practical utility.
I believe that relying solely on task-incremental experiments is insufficient to establish meaningful results.

**Questions:**

- Is the Hebbian rule a better fit for spiking neural networks compared to other approximate PCA approaches?
- If it is, what makes the Hebbian rule more suitable, and why aren't the other approaches as effective?

---

> ### Author Response · Authors · 2023-11-19
> **Response to Reviewer braR**
>
> Thank you for your valuable comments. We respond to your comments and questions as follows.
>
> 1. About Hebbian learning and other approximations of PCA.
>
> It is true that there are many PCA approximation algorithms. However, a basic focus of our work is neuromorphic computing, and the biologically inspired Hebbian learning well fits neuronal computation.
>
> We first introduce more background on neuromorphic computing. Neuromorphic computing [1] aims at the computation inspired by the structure and function of human brains. At the hardware level, neuromorphic chips are designed to imitate biological neurons, such as their spiking property and synaptic connections with local storage of weights for in-memory computation, for highly energy-efficient and parallel event-driven computation with avoidance of frequent memory transpose (the computation architecture is different from the commonly used hardware with von Neumann architecture such as CPU or GPU). At the algorithm level, we are interested in developing methods compatible with properties and operations of neurons so that they are possible for deployment on hardware. As neuromorphic hardware is under development considering software-hardware co-design, most algorithms are simulated on common computational devices while considering neuromorphic properties.
>
> For PCA algorithms, most of them require a lot of operations not corresponding to neuronal activities and connections, and the learning/update of the matrix does not follow the (biological) learning rule that may be designed for neuromorphic computing considering the locality property.
>
> On the other hand, biologically inspired Hebbian learning well fits neural computation, and can also achieve sota convergence rate in streaming PCA [2]. From the perspective of general computation, neuroscience-backed algorithms will indeed boil down to matrix-vector arithmetic as other methods. But considering neuromorphic computing, Hebbian learning can be mapped to the computation of neurons and synapses and implemented on neuromorphic hardware, while other methods do not. So Hebbian learning is more suitable for SNNs.
>
> Additionally, Hebbian learning enables unbiased streaming PCA from large data for better subspace construction than PCA methods adopted by previous works. This also has improvement and leads to better performance.
>
> 2. About the task-incremental setting.
>
> We would like to clarify that we consider both task-incremental and domain-incremental settings, as what previous gradient projection methods do. Continual learning is classified into three fundamental types: task-incremental, domain-incremental, and class-incremental [3]. Task-CL requires task id at training and testing, domain-CL shares the whole network and does not require task id, while class-CL requires comparing new tasks and old tasks and inferring context without task id. Many previous works call both task- and domain-incremental as task-incremental, and here we distinguish them as in [3] to emphasize that our method does not necessarily need task id. Following experimental setups of previous gradient projection methods [4,5], we consider both task- (CIFAR-100, miniImageNet, 5-Dataets) and domain-CL (PMNIST) settings. As for class-CL, as discussed in Related Work, it inevitably requires some kind of replay of old experiences for good performance since it expects explicit comparison between new classes and old ones [6,4]. So we also do not focus on it and it can be future work to study if HLOP can be combined with some replay methods or context-dependent processing modules similar to biological systems [7] (e.g., with a task classifier) for better class-incremental tasks.
>
> Our PMNIST experiment in Table 2 is domain-CL that does not require task id, since the classifier is shared. Here we further supplement the domain-incremental setting results on the larger 5-Datasets with a larger network (reduced ResNet-18) to show the scalability. The comparison results are below (where HAT requires task id and is not feasible) and the results also show the superior performance of HLOP.
>
> | Method | Domain-CL 5-Datasets (ACC, BWT) |
> | :----: | :----: |
> | *Multitask* | *82.81*, / |
> | Baseline | 32.46, -74.86 |
> | MR | 70.26, -24.12 |
> | EWC | 35.80, -65.18 |
> | HAT | N.A., N.A. |
> | GPM | 45.62, -56.28 |
> | **HLOP (ours)** | **79.59**, **-9.94** |
>
> [1] Towards spike-based machine intelligence with neuromorphic computing. Nature, 2019.
>
> [2] ODE-inspired analysis for the biological version of Oja’s rule in solving streaming PCA. COLT, 2020.
>
> [3] Three types of incremental learning. Nature Machine Intelligence, 2022.
>
> [4] Gradient projection memory for continual learning. ICLR, 2021.
>
> [5] TRGP: Trust region gradient projection for continual learning. ICLR, 2022.
>
> [6] Brain-inspired replay for continual learning with artificial neural networks. Nature Communications, 2020.
>
> [7] Biological underpinnings for lifelong learning machines. Nature Machine Intelligence, 2022.

---

> > ### Comment · Reviewer_braR · 2023-11-20
> >
> > Thank you for the response. Unfortunately, however, my primary concern about justifying Hebbian learning is unresolved.
> > In the review, I asked the authors to justify the claim that "the Hebbian approach is particularly suitable for spiking neural networks."
> > In the authors' response, the only relevant parts are:
> > > biologically inspired Hebbian learning well fits neural computation
> >
> > > Hebbian learning can be mapped to the computation of neurons and synapses and implemented on neuromorphic hardware, while other methods do not
> >
> > These are just repetitions of the claim, while I requested evidence for it.
> > I'm lowering the soundness and contribution scores accordingly.

---

> > > ### Author Response · Authors · 2023-11-20
> > >
> > > Thank you for your further question. Sorry that we may not explain the background clearly. We emphasize that **spiking neural networks** belong to **neuromorphic computing** [1] that we give a more detailed introduction in our response, and studying SNNs algorithms eventually aims at combination with neuromorphic hardware that imitate human brains (because SNNs are imitating biological neurons and they fit these hardware more rather than common hardware like GPU – on GPUs, SNNs do not have any energy advantage). So whether an algorithm is particularly suitable for SNNs can be justified by its fitness to neuromorphic computing. All our response is *relevant* to your question, as we explain what properties neuromorphic computing should consider and why. Particularly, neuromorphic computing for SNNs requires mapping computation to neural activities and synapses which can be realized by in-memory computation on hardware (it is not general computation hardware with arbitrary matrix-vector arithmetic, in order to imitate the energy efficiency of brains). Hebbian learning conforms to neuromorphic properties (because itself is inspired by biological learning rule, as SNNs are inspired by biological neurons) and is particularly suitable for neuromorphic computing of SNN.
> > >
> > > Additionally, as we responded to Reviewer Me3Q, we compare the difference between ANNs and SNNs with different continual learning methods, and our method has more improvements over others for SNNs particularly on 5-Datasets (the results are also shown below). It may imply that our method can be better combined with spikes, for example probably because subspaces expanded by spike signals are harder to learn and therefore our Hebbian learning performing streaming PCA has more advantages over GPM that only performs SVD/PCA on a small batch of data which can be biased. It can be an empirical evidence that our method is more suitable for SNNs.
> > >
> > > | Neural Network | Method | PMNIST (ACC, BWT) | CIFAR-100 (ACC, BWT) | miniImageNet (ACC, BWT) | 5-Datasets (ACC, BWT) |
> > > | :----: | :----: | :----: | :----: | :----: | :----: |
> > > | SNN | *Multitask* | *96.15*, / | *79.70*, / | *79.05*, / | *89.67*, / |
> > > | SNN | Baseline | 70.61, -28.88 | 75.13, -4.33 | 40.56, -32.42 | 45.12, -60.12 |
> > > | SNN | MR | 92.53, -4.73 | 77.67, -1.01 | 58.15, -8.71 | 79.66, -14.61 |
> > > | SNN | EWC | 91.45, -3.20 | 73.75, -4.89 | 47.29, -26.77 | 57.06, -44.55 |
> > > | SNN | HAT | N.A., N.A. | 73.67, -0.13 | 50.11, -7.63 | 72.72, -22.90 |
> > > | SNN | GPM | 94.80, -1.62 | 77.48, -1.37 | 63.07, -2.57 | 79.70, -15.52 |
> > > | SNN | **HLOP (ours)** | 95.15, -1.30 | 78.58, -0.26 | 63.40, -0.48 | 88.65, -3.71 |
> > > | ANN | *Multitask* | *96.81*, / | *80.33*, / | *82.05*, / | *88.65*, / |
> > > | ANN | Baseline | 73.55, -25.67 | 71.47, -7.82 | 38.87, -31.58 | 68.54, -30.32 |
> > > | ANN | MR | 92.44, -5.00 | 75.59, -2.68 | 55.15, -6.79 | 83.98, -9.29 |
> > > | ANN | EWC | 90.16, -3.46 | 72.49, -6.01 | 47.41, -21.46 | 68.60, -29.90 |
> > > | ANN | HAT | N.A., N.A. | 71.31, 0.00 | 56.98, -1.64 | 88.40, -3.63 |
> > > | ANN | GPM | 94.92, -1.56 | 77.56, -1.42 | 65.13, -0.96 | 84.88, -9.17 |
> > > | ANN | **HLOP (ours)** | 95.25, -1.28 | 78.98, -0.34 | 66.80, 1.93 | 88.68, -2.74 |
> > >
> > > [1] Towards spike-based machine intelligence with neuromorphic computing. Nature, 2019.

---

> > > > ### Author Response · Authors · 2023-11-20
> > > >
> > > > Additionally, not sure about what specific “evidence” you expect, we supplement some references on neuromorphic hardware. Intel’s Loihi [1] is one of the most famous neuromorphic chips for SNNs with on-chip learning abilities, and it only supports Hebbian-type learning rules or with a third reward modulator. According to the formulation in [1], its update rule can be programmed as some products of local neuronal information, such as pre-/post-synaptic spike count, traces, etc., or with a reward signal. Our Hebbian learning can theoretically be supported by the hardware, while other PCA approximation with general computation of matrix-vector arithmetic cannot. As discussed in our previous response, neuromorphic computing and hardware of SNNs mainly follow brain functions and have some properties such as locality for learning rules. So biologically inspired Hebbian learning is more interesting and suitable to SNNs. We answer your question from the perspective of basic neuromorphic computing properties, evidence from neuromorphic hardware, as well as empirical performance results between ANNs and SNNs. Could you please specify what specific “evidence” you want? We would like to know if our responses have answered your question.
> > > >
> > > > [1] Davies et al. Loihi: A neuromorphic manycore processor with on-chip learning. IEEE Micro, 2018.

---

> > > > > ### Comment · Reviewer_braR · 2023-11-21
> > > > >
> > > > > Thank you for more detailed comments.
> > > > > I wanted a more rigorous and scientific argument than "both Hebbian learning and SNNs are inspired by biological neurons," which is just an analogy.
> > > > >
> > > > > The last response provides an argument that is much closer to what I anticipated:
> > > > > > Intel’s Loihi [1] is one of the most famous neuromorphic chips for SNNs with on-chip learning abilities, and it only supports Hebbian-type learning rules or with a third reward modulator.
> > > > >
> > > > > And there is a more precise description I found in [1]:
> > > > > >SNN synaptic weight adaptation rules must satisfy a **locality constraint**: each weight can only be accessed and modified by the destination neuron, and the rule can only use locally available information, such as the spike trains from the presynaptic (source) and postsynaptic (destination) neurons.
> > > > >
> > > > >
> > > > > After reading this, however, I am more puzzled.
> > > > > If neuromorphic chips require a learning algorithm that fulfills the locality constraint, it cannot handle regular back-propagation, which is part of the proposed method.
> > > > > I looked into the PyTorch code in the supplementary material, and there surely is the standard backpropagation code, which would not be runnable on neuromorphic chips.
> > > > > ```
> > > > > loss.backward()
> > > > > optimizer.step()
> > > > > ```
> > > > >
> > > > > Given that the proposed method has not been tested on actual neuromorphic hardware, I am skeptical about its compatibility with neuromorphic chips.
> > > > > To counter my argument, demonstrating the proposed approach on real neuromorphic hardware would be the most effective.
> > > > > I'm open to correction if there's any misunderstanding on my part.

---

> > > > > > ### Author Response · Authors · 2023-11-21
> > > > > >
> > > > > > Thank you for your further question. We respond to your question from several points.
> > > > > >
> > > > > > 1. First, Loihi [1] supports Hebbian-type learning rules or with a third reward modulator, and the third factor can act as a global (reward/error) signal (see the fourth point below the sentence you quoted from [1], and Eq. (4) as well as Table 1 in [1]). This is from the three-factor Hebbian learning proposed and verified in neuroscience [2,3,4,5]. In neuroscience, local Hebbian learning is first observed and proposed as synaptic plasticity, i.e., neurons wire together if they fire together [6], while later, it is found that neuromodulators (such as dopaminergic system) influence synaptic plasticity as reward or penalty, which supports some kind of global learning. The locality constraint mainly describes original Hebbian-type learning, and a third factor modulating this synaptic plasticity is also supported. Loihi should support flexible rules with local information and a global modulator [1].
> > > > > >
> > > > > > 2. Second, with a third factor, the error from backpropagation can be represented in this form, while the backpropagation procedure has some problem known as the “weight transport” problem [7]. That’s why in Section 5.2, we verified different error propagation methods including biologically more plausible alternatives feedback alignment (FA) [7] and sign symmetric (SS) [8]. In the Technology Brief of the latest Loihi 2 [9], it is said that it provides support for many of the latest neuro-inspired learning algorithms under study, including approximations of the error backpropagation algorithm.
> > > > > >
> > > > > > 3. Third, we would like to emphasize that our method is orthogonal to normal SNN training methods (such as with BP or FA /SS) and is flexible to utilize arbitrary training methods based on presynaptic activities/traces (since we only modify activity traces no matter how error signal is propagated), and we verified this in Sections 5.1 and 5.2. Our method can be viewed as a flexible module to be combined with normal training methods, not restricted by backpropagation.
> > > > > >
> > > > > > 4. Fourth, we would like to mention that neuromorphic hardware is still under development and not generally accessible, and it is also considering software-hardware co-design [10], so most algorithms are simulated on common computational devices [11,12] while considering neuromorphic properties. We currently have no access to neuromorphic chips and follow previous works to simulate on GPUs.
> > > > > >
> > > > > > 5. Fifth, we would like to mention that while neuromorphic hardware is under development, we should consider and investigate algorithms consistent with neuromorphic properties because hardware is and will be designed following them, serving as non-von Neumann architectures composed of neurons and synapses [10]. Neuromorphic computing is developed to break the separation of CPUs and memory units in von Neumann architectures, which requires frequent memory exchange that consumes a considerable amount of energy compared with the compute energy. Instead, imitating human brains that compute with locally stored weights as well as event-driven computation, neuromorphic chips target high energy efficiency [10]. And that’s a reason behind why Hebbian-type learning with locality constraint is more interesting to SNNs. Also, it is not general computer supporting arbitrary matrix-vector arithmetic, so mapping computation to neurons and synapses like Hebbian learning is more suitable than other PCA approximations.
> > > > > >
> > > > > > [2] Neuromodulated spike-timing-dependent plasticity, and theory of three-factor learning rules. Frontiers in Neural Circuits, 2016.
> > > > > >
> > > > > > [3] Learning with three factors: modulating Hebbian plasticity with errors. Current Opinion in Neurobiology, 2017.
> > > > > >
> > > > > > [4] Eligibility traces and plasticity on behavioral time scales: experimental support of neohebbian three-factor learning rules. Frontiers in Neural Circuits, 2018.
> > > > > >
> > > > > > [5] Control of synaptic plasticity in deep cortical networks. Nature Reviews Neuroscience, 2018.
> > > > > >
> > > > > > [6] The organization of behavior: A neuropsychological theory. Psychology Press, 2005.
> > > > > >
> > > > > > [7] Random synaptic feedback weights support error backpropagation for deep learning. Nature Communications, 2016.
> > > > > >
> > > > > > [8] Biologically-plausible learning algorithms can scale to large datasets. ICLR, 2018.
> > > > > >
> > > > > > [9] Taking neuromorphic computing to the next level with Loihi 2. 2021.
> > > > > >
> > > > > > [10] Opportunities for neuromorphic computing algorithms and applications. Nature Computational Science, 2022.
> > > > > >
> > > > > > [11] A solution to the learning dilemma for recurrent networks of spiking neurons. Nature Communications, 2020.
> > > > > >
> > > > > > [12] Online training through time for spiking neural networks. NeurIPS, 2022.

---

> > > > > > > ### Comment · Reviewer_braR · 2023-11-22
> > > > > > >
> > > > > > > Thank you for your response.
> > > > > > > It addressed most of my concerns, and as a result, I'm adjusting my score accordingly and recommending acceptance.
> > > > > > > Please understand that I misunderstood several key concepts due to my limited expertise in SNNs and neuromorphic hardware.
> > > > > > > It was a pleasure to engage in an insightful discussion.

---

### Official Review · Reviewer_oRoY · 2023-10-31

**Soundness:** 3 good
**Presentation:** 3 good
**Contribution:** 4 excellent
**Rating:** 6
**Confidence:** 3

**Summary:**

Unlike biological intelligence, current deep learning suffers from catastrophic forgetting— upon learning new tasks, networks often lose the ability to solve previously learned tasks.
One set of methods to solve catastrophic forgetting is orthogonal gradient projection, in which the learning gradient for new tasks are projected to a subspace that is approximately orthogonal to the subspace of the gradient of the network w.r.t. the weights for old tasks.
However, they are not applicable to Spiking Neural Networks (SNNs), which is the predominant architecture for neuromorphic learning.
This paper proposes Hebbian Learning based Orthogonal Projection, or HLOP, an orthogonal gradient projection method that extracts principal subspaces of neuronal activities using Hebbian learning.
HLOP is compatible with SNNs, and outperforms other continuous learning methods on several computer vision datasets.
Thus, it may be a promising new direction for continual neuromorphic learning.

**Strengths:**

- The authors introduce a novel combination of Hebbian learning with orthogonal projection to solve catastrophic forgetting for SNNs. While using Hebbian learning to find principal subspaces is not a new technique, its application to this problem is both novel and elegant.
- HLOP achieves strong empirical performance, surpassing all other continual learning methods that the authors benchmarked against.

**Weaknesses:**

- The final paragraph of the intro that discusses your contributions is a bit dense. I suggest breaking it up, and allocating more intro space to discuss your contributions, as it’s the most important part of your intro. In particular, I think you should have a separate paragraph to discuss the experimental setup and results, and include some performance numbers to quantify the strength of your method; currently its strength is hard to judge from the intro.

- The background on SNNs provided in Section 3.1 is a word-for-word replica of Section 3.1 in Xiao et al. (2022).
It’s fine to reuse definitions from previous work, but a direct replica like this should be explicitly attributed to avoid plagiarism concerns (even if it’s your own work).

- There is almost no discussion on the current limitations/challenges of HLOP and the experimental design, making it hard to judge the tradeoffs between using HLOP versus other approaches.
I’d like to see the authors also include a discussion on the potential downsides of HLOP and limitations of the evaluation process.

- Minor issue, but the sentence in Section 5.2 explaining the weight transport problem of backprop is very long and hard to parse.
I suggest either rewriting it and breaking it down to smaller chunks, or removing it entirely as it’s not central to your work; just stating that FA and SS are more biologically-plausible and more amenable to neuromorphic hardware is enough.

**Questions:**

- At the end of Section 3.2, you mention that previous methods “cannot be implemented by neuronal operations for neuromorphic computing,” but do not provide further explanation.
To me, this is important because it provides critical motivation for your method; it highlights why your novel approach using Hebbian learning is required.
I understand that these methods cannot be directly implemented on SNNs out-of-the-box, but can you elaborate on why it is difficult or infeasible to adapt them for SNNs?

- The performance of HLOP is close to the upper bound (specified by Multitask performance) for each dataset except miniImageNet, where there is a large gap. Can you provide reasoning or intuition for why this is the case?

- Your results show that HLOP outperforms several other continual learning methods. Can I assume that the results achieved with HLOP are state-of-the-art on these datasets? Or are there other methods that you did not compare against?

---

> ### Author Response · Authors · 2023-11-19
> **Response to Reviewer oRoY (Part 1/2)**
>
> Thank you for appreciating our work and providing valuable comments. We respond to your comments and questions as follows.
>
> 1. Writing issues.
>
> Thank you for your kind advice. Following your suggestions, we reorganize/modify descriptions of the final paragraph of the intro as well as Section 3.1, and move the sentence in Section 5.2 to the footnote. Please check the updated version.
>
> 2. Limitations/challenges of HLOP and evaluation.
>
> As for our method, HLOP introduces additional computational costs for learning the lateral weights of each layer during training, which will increase training costs in our current implementation codes. While this process can theoretically be parallel to the normal forward-backward propagation of network training as discussed in the paper, such parallelization may not be easily realized in established deep learning libraries like PyTorch. So for our implementation of GPU training, the training time would be slightly longer. It can be future work on low-level code optimization or consideration of parallel neuromorphic computing hardware. Additionally, HLOP currently requires a manual specification for the number of subspace neurons, which may be improved for automatic and adaptive allocation. For example, the Generalized Hebbian Algorithm can perform Gram-Schmidt orthonormalization on the rows of weight matrices, which may help to sweep out unnecessary neurons and connections, but it requires some non-local information and may not be directly suitable to our method. It can be interesting for future work to study improvements.
>
> As for the evaluation, similar to other gradient projection methods, this work mainly focuses on task-incremental and domain-incremental continual learning settings. There is another class-incremental setting which, as discussed in Related Work, inevitably requires some kind of replay of old experiences for good performance since it expects explicit comparison between new classes and old ones [1,2]. So following previous works, our evaluation mainly focuses on task- and domain-incremental settings. It can be future work to study if HLOP can be combined with some replay methods or context-dependent processing modules similar to biological systems [3] (e.g., with a task classifier) for better class-incremental tasks.
>
> We have added the above discussions in Appendix F in the revised version.
>
> 3. Why previous methods cannot be implemented by neuronal operations for neuromorphic computing.
>
> We would like to first introduce more background on neuromorphic computing. Neuromorphic computing aims at the computation inspired by the structure and function of the human brain. At the hardware level, neuromorphic chips are designed to imitate biological neurons, such as their spiking property and synaptic connections with local storage of weights for in-memory computation, for highly energy-efficient and parallel event-driven computation with avoidance of frequent memory transpose (the computation architecture is different from the commonly used hardware with von Neumann architecture such as CPU or GPU). At the algorithm level, we are interested in developing methods compatible with some properties and operations of neurons so that they are possible for deployment on the hardware. Also, note that neuromorphic hardware is under development considering software-hardware co-design, so most algorithms are simulated on common computational devices while considering neuromorphic properties.
>
> When it comes to previous methods mentioned in Section 3.2, they do not consider neuromorphic properties and require complex computation beyond neuronal operations. First, for calculation of the projection matrix, they require operations such as SVD on a batch of data or calculating inverses of matrices, which needs a lot of (general) operations not corresponding to neuronal activities and connections, and the learning/update of the projection matrix does not follow the (biological) learning rule that may be designed for neuromorphic computing considering the locality property. Instead, we propose to leverage biologically inspired Hebbian learning which can also be directly integrated into the lateral circuit. Second, they do not consider how projection can be implemented. They require matrix multiplication of a projection matrix with weight gradients, which cannot correspond to operations in neurons and synaptic connections. While neurons can perform matrix multiplication considering neuronal activations and local connection weights, the above calculation cannot be directly mapped to such computation and it is unclear how the projection matrix can be locally stored. Instead, we propose a lateral neural circuit to modify the presynaptic activity traces for projection. The previous methods can be implemented on common devices as in our experiments, but considering neuromorphic computing, they are difficult to realize and we seek methods that have more similarity to biological neurons.

---

> > ### Author Response · Authors · 2023-11-19
> > **Response to Reviewer oRoY (Part 2/2)**
> >
> > 4. About performance on miniImageNet.
> >
> > In previous works, continual learning results on miniImageNet also have a large gap to Multitask compared with other datasets [2,4]. This is probably because different tasks in this dataset have correlations and can improve each other. Multitask learning encourages knowledge transfer between tasks, while most continual learning methods mainly focus on solving catastrophic forgetting when learning tasks successively and cannot leverage this property. It is another important topic on how to better support knowledge transfer between tasks.
> >
> > 5. About other methods.
> >
> > Not many continual learning methods consider SNNs, and we do not find SNN baselines on these common datasets. As we cannot exhaustingly reimplement all continual learning methods designed for ANNs, we mainly implement and compare representative methods of different kinds of methods (i.e., memory replay, regularization, and gradient projection) based on their released codes under our SNN settings. Note that the performance of some methods can differ considering ANNs and SNNs (please see our response to Reviewer Me3Q). Our results are state-of-the-art for SNNs.
> >
> > There are some recent works [4] improving the gradient projection method GPM [2], which is our state-of-the-art baseline. We attempt to test TRGP [4] based on their released code under our setting. On PMNIST and CIFAR-100, it does not outperform GPM as shown below, while on miniImageNet and 5-Datasets, we encounter strange failures that the training fails after one or two tasks (i.e., the performance for new tasks is only random guess, but the performance for old tasks is maintained). It may probably be the difference between ANN and SNN, or hyperparameters and training settings, or some bugs in the code. Given the limited time, we are unable to thoroughly analyze their method and codes. We think that our current comparison is enough to show the superiority of our method.
> >
> > | Method | PMNIST (ACC, BWT) | CIFAR-100 (ACC, BWT) |
> > | :----: | :----: | :----: |
> > | GPM | 94.80, -1.62 | 77.48, -1.37 |
> > | TRGP | 94.56, -1.08 | 76.77, -2.61 |
> > | **HLOP (ours)** | 95.15, -1.30 | 78.58, -0.26 |
> >
> > [1] van de Ven et al. Brain-inspired replay for continual learning with artificial neural networks. Nature Communications, 2020.
> >
> > [2] Saha et al. Gradient projection memory for continual learning. ICLR, 2021.
> >
> > [3] Kudithipude et al. Biological underpinnings for lifelong learning machines. Nature Machine Intelligence, 2022.
> >
> > [4] Lin et al. TRGP: Trust region gradient projection for continual learning. ICLR, 2022.

---

### Author Response · Authors · 2023-11-19
**A summary of paper updates**

We thank all reviewers for their valuable comments and feedback. We have uploaded the updated version of our paper based on the reviews. Revisions are marked as blue in the text. The updates are summarized as follows:

1. Following suggestions from Reviewer oRoY and Reviewer Me3Q, we reorganize/modify descriptions of the final paragraph of the intro, Section 3.1, and some sentences, and modify the boldface in Table 2.

2. In response to Reviewer oRoY, we supplement some discussions of limitations/challenges in Appendix F.

3. In response to Reviewer oRoY and Reviewer braR, we supplement more introduction and discussions on neuromorphic computing in Appendix F.

4. In response to Reviewer braR, we clarify the considered task-incremental and domain-incremental settings in the paper, and supplement the domain-CL results on 5-Datasets in Appendix G.

5. In response to Reviewer Me3Q, we supplement all the corresponding results of ANNs as well as discussions in Appendix G.

Detailed responses to every question are in each separate response to reviewers.

---

### Public Comment · ~Fan_Wang4 · 2024-06-28
**Missing References**

Very nice work, but we hope the following paper could be added to reference since it is very relevant.
Evolving Decomposed Plasticity Rules for InformationBottlenecked Meta-Learning
https://arxiv.org/abs/2109.03554

---

### Meta-Review · Area_Chair_ZXvj · 2023-12-04

**Metareview:**

This paper studies continual learning in spiking neural networks (SNNs). The authors develop a new technique to make synaptic weight updates in orthogonal subspaces, such that previously learned abilities are not interfered with by new learning. The system relies on a mixture of lateral connectivity, Hebbian, and anti-Hebbian learning. The authors show using a variety of datasets that they're system can avoid catastrophic forgetting better than previous approaches.

The initial reviews for this paper were mixed, and this was a borderline paper, with a variety of concerns raised, including concerns about scalability and whether the design decisions were well-justified. In particular, there were concerns from one of the reviewers that the paper did a poor job justifying the use of Hebbian and anti-Hebbian updates in SNNs and comparing to previous methods for PCA in neural networks. But, after fairly extensive discussion with the authors, and some discussion amongst the reviewers, the negative reviewers raised their scores above the acceptance threshold, and thus, an 'accept' decision was reached.

**Justification For Why Not Higher Score:**

Given that this was a borderline paper, I don't think a spotlight is warranted.

**Justification For Why Not Lower Score:**

All of the reviews were above the accept threshold by the end, and I didn't see any reason to contradict the reviewers here.

---

### Decision · Program_Chairs · 2024-01-16

Accept (poster)